# Sea level regulated tetrapod diversity dynamics through the Jurassic/Cretaceous interval

Jonathan P. Tennant[1], Philip D. Mannion[1] & Paul Upchurch[2]

Reconstructing deep time trends in biodiversity remains a central goal for palaeobiologists, but our understanding of the magnitude and tempo of extinctions and radiations is confounded by uneven sampling of the fossil record. In particular, the Jurassic/Cretaceous (J/K) boundary, 145 million years ago, remains poorly understood, despite an apparent minor extinction and the radiation of numerous important clades. Here we apply a rigorous subsampling approach to a comprehensive tetrapod fossil occurrence data set to assess the group's macroevolutionary dynamics through the J/K transition. Although much of the signal is exclusively European, almost every higher tetrapod group was affected by a substantial decline across the boundary, culminating in the extinction of several important clades and the ecological release and radiation of numerous modern tetrapod groups. Variation in eustatic sea level was the primary driver of these patterns, controlling biodiversity through availability of shallow marine environments and via allopatric speciation on land.

[1] Department of Earth Science and Engineering, Imperial College London, London SW6 2AZ, UK. [2] Department of Earth Sciences, University College London, Gower Street, London WC1E 6BT, UK. Correspondence and requests for materials should be addressed to J.P.T. (email: jon.tennant.2@gmail.com).

The Phanerozoic era records evidence for numerous large-scale extinction events, varying from catastrophic mass extinctions to phases of high ecological turnover[1,2]. Among these events, the Jurassic/Cretaceous (J/K) boundary, 145 Myr ago, was originally considered to be one of eight major extinctions[3], but taxonomic selectivity and an apparent geographic constraint to Europe[4,5] mean that it is now more commonly regarded as a minor extinction event. This perceived lack of extinction intensity, coupled with the apparent absence of marked environmental perturbations and loss of major clades, has led to the J/K boundary remaining the most poorly understood of major Mesozoic stratigraphic boundaries in terms of the links between abiotic and biotic patterns[6], despite an estimated extinction of up to 20% of genera[2,7].

Early investigations of palaeobiodiversity were based on a literal 'raw' reading of the fossil record[2,8] that did not explicitly account for how structural modifications in the architecture of the geological record influence the availability of fossils for sampling, or how we have sampled from this incomplete record. Such macrostratigraphic and anthropogenic variations are often termed 'sampling biases', which determines the underlying artefacts that affect our understanding of diversity in deep time. In addition to this issue are those of 'common cause' and 'redundancy'[9–11]. The former refers to an external factor that drives both sampling and diversity, such as climate or sea level, and the latter phenomenon arises from non-independence of sampling and diversity metrics[10,12,13]. These three phenomena account for the different ways in which sampling and diversity can covary, and imply that a raw reading of the fossil record can often be inappropriate for reconstructing past diversity patterns. Teasing apart hypotheses of sampling bias, redundancy, common cause and their impact on palaeobiodiversity remains a problematic but fundamental goal of current palaeobiological research[11,12,14], particularly in the terrestrial realm.

Recently, our understanding of the diversity dynamics of major tetrapod clades has increased through the use of quantitative techniques that attempt to mitigate the impact of uneven sampling on raw diversity patterns. Our revised view hints at a previously obscured macroevolutionary complexity through the J/K interval that includes multiple independent marine radiations, as well as the origins and extinctions of many important clades[6,15–19]. Together, these independent studies suggest that there was a sharp decrease in the diversity of dinosaurs[12,16,20], pterosaurs[21], crocodyliforms[18] and marine reptiles[15,22] across the J/K boundary. Combined with this faunal loss is evidence for pulses of ecological turnover[6,17,19] and the subsequent Early Cretaceous origin of numerous extant clades, including marine turtles[23], eusuchian crocodylomorphs[24] and several squamate groups[25], which might or might not be related to J/K boundary events. For example, nearly all non-pterodactyloid pterosaurs went extinct at the boundary, before the apparent diversification of pterodactyloids during the Early Cretaceous[21]. Medium- and large-sized saurischian dinosaurs appear to have been selectively targeted by extinction[26], preceding the apparent Early Cretaceous radiations of larger- and smaller-bodied diverse theropod clades[16,27], including birds[28]. The extinction of many basal crocodyliforms across the J/K boundary[18] is followed by rapidly increasing diversification rates in notosuchians and eusuchians[24], although the relative timing and magnitude of all of these diversification events is based on a superficial reading of the fossil record, and obscured by incomplete lineage sampling. Evidence for this seemingly broad pattern of decline contrasts with the relatively high lineage survivability documented in sauropterygians[17], metriorhynchoid crocodylomorphs[29] and ichthyosaurs[30,31], and an increase in diversity in non-marine turtles[19] and mammaliaforms[32] across the J/K transition. This inconsistent pattern of diversity and extinction between marine and non-marine groups suggests that instead of a single, underlying mechanism, different processes governed diversity patterns between environments.

Irrespective of this emerging picture, there is currently considerable disagreement over the taxonomic inclusivity, magnitude and timing of any putative J/K boundary extinction[6,17,26,33]. To address this issue, we have assembled one of the largest tetrapod occurrence data sets to date, comprising over 12,000 individual body fossil occurrences, representing more than 2,000 genera (Supplementary Data 1 and Supplementary Figs 1 and 2). We use both a raw empirical approach and apply a rigorous subsampling protocol, Shareholder Quorum Subsampling (SQS)[5,34], to reconstruct palaeobiodiversity. In addition, we estimate phylogenetic diversity for groups in which there are well-sampled global phylogenies available, and also calculate global extinction and origination rates, an aspect that has not been included in most recent assessments of tetrapod palaeobiodiversity. Maximum likelihood is used to fit models that describe both sampling and extrinsic environmental parameters to resulting diversity curves (Supplementary Data 2), allowing us to assess the primary drivers of tetrapod macroevolutionary patterns through the J/K boundary. Here we (1) quantify the magnitude of extinction and diversity loss of higher tetrapod clades across the J/K transition, and refine the timing of these events; (2) evaluate the impact of geological sampling biases; (3) assess the potential environmental drivers of resulting patterns; and (4) place such patterns in the ecological context of the radiation and extinction of major clades. To our knowledge, this is the first time that the relationship between both geological and environmental factors and standardized diversity has been explored on the scale of all major tetrapod groups.

## Results

**Uncorrected taxonomic diversity**. 'Raw' patterns of uncorrected, global genus-level diversity show a decline in almost all taxonomic groups across the J/K boundary (Supplementary Data 3). Marine tetrapod diversity was consistently almost an order of magnitude lower than non-marine diversity (Fig. 1), and shows an ~75% decline across the J/K boundary. In non-marine faunas, small-bodied taxa (lepidosauromorphs, lissamphibians and mammals) exhibited either flat diversity or a small increase across the boundary, whereas medium- to large-bodied groups, including pterosaurs, crocodyliforms, turtles and all three major dinosaur groups, reveal differential patterns of decline from 33 to 80% losses of standing diversity.

**Global subsampled and phylogenetic diversity**. Subsampled diversity estimates show a more nuanced pattern than the raw

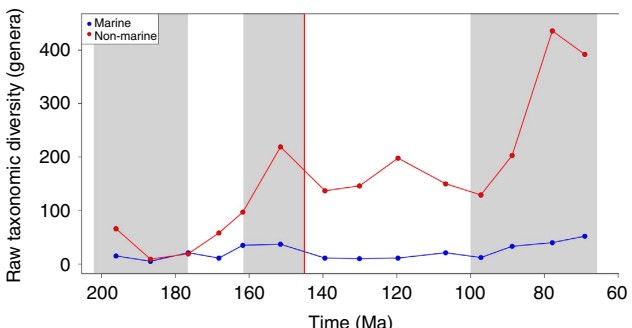

**Figure 1 | Raw taxonomic diversity.** Taxonomic diversity estimate (TDE) for marine and non-marine Jurassic and Cretaceous tetrapods, grouped into approximately equal 10 million year time bins.

taxonomic estimates (results reported using a quorum of 0.4; Supplementary Methods). Using 10 million year time bins on a global scale, the only non-marine group to show a notable drop in diversity across the J/K boundary is Crocodyliformes[18] (including Thalattosuchia—see Tennant *et al.*[35] for further discussion). Mammals and lissamphibians appear to have increased in diversity, with the former almost doubling, in marked contrast to previous studies documenting only a small increase[32]. Theropod and ornithischian dinosaurs both show a small increase in diversity across the J/K boundary, contrasting with results obtained using a residual diversity approach[16], but we were unable to recover a signal for sauropods in the earliest Cretaceous. Choristoderes and Aves are too poorly sampled throughout this interval to identify any diversity patterns with confidence. Marine crocodyliforms also suffered a decline across the J/K boundary[18], in contrast with a slight increase in sauropterygians[17]. The dynamics of ichthyosaur diversity remain obscure. However, at this relatively coarse resolution, it is difficult to distinguish whether these patterns occurred through the J/K boundary (that is, the Tithonian–Berriasian), or represent the lumping together of discrete signals from different time bins (that is, the Kimmeridgian + Tithonian and Berriasian + Valanginian) (Supplementary Tables 1 and 2).

At a finer geological stage level, a markedly different global pattern emerges. This results from the differences in the size and shape of the sample pool (that is, the taxonomic abundance distribution) due to variation in the duration of our time bins. For the larger-scale bootstrapped SQS analyses at the tetrapod level for marine and non-marine taxa (Supplementary Fig. 3), we see that non-marine tetrapods show an uncertain diversity pattern through the J/K boundary. Differences between the upper and lower confidence intervals at this level suggest that there might even have been a slight increase in non-marine tetrapod diversity. There is a definitive crash in diversity in the Hauterivian, with very narrow confidence intervals. In marine tetrapods, the overall pattern of decline through the J/K transition remains distinct. There is a continuous decline from the Tithonian to the Hauterivian, a pattern that remains constrained within the confidence intervals. However, these patterns fail to account for, or detect, the smaller-scale variations we find at the finer clade level. Note also that these groups do not represent clades, but ecological groups.

In non-marine faunas, ornithischians and theropods show declines of around 33% and 75% diversity loss, respectively (Fig. 2a), from the Tithonian to the Berriasian. For ornithischians, this result is similar to that recovered from residual diversity estimates[16], and reflects the decline of stegosaurs. However, we cannot rule out that subsampled ornithischian diversity actually increases when applying a bootstrapping sensitivity test (Supplementary Fig. 3). This signal for theropods is highly distinct, but remains when we apply bootstrapping, suggesting that the decline might have been even more severe across the J/K boundary. Previous results based on residual diversity estimates show either a steady increase (collections-based) or small decline (formation-based) in theropod diversity through the J/K boundary[16] (Supplementary Methods), instead of the more prominent decline we recover here. Sauropods are too poorly sampled in the Berriasian to reveal a signal, but their Valanginian diversity was only 37% of their Tithonian diversity, reflecting the decline of non-neosauropods, diplodocids and basal macronarians[36], representing a substantial loss around the J/K transition. Our results partly support the marked decline in sauropod diversity recovered using residual diversity estimates[16], before the Barremian radiation of titanosauriforms[36]. Furthermore, our phylogenetic diversity estimates (PDEs) for each of the three major dinosaur clades provide some support for our

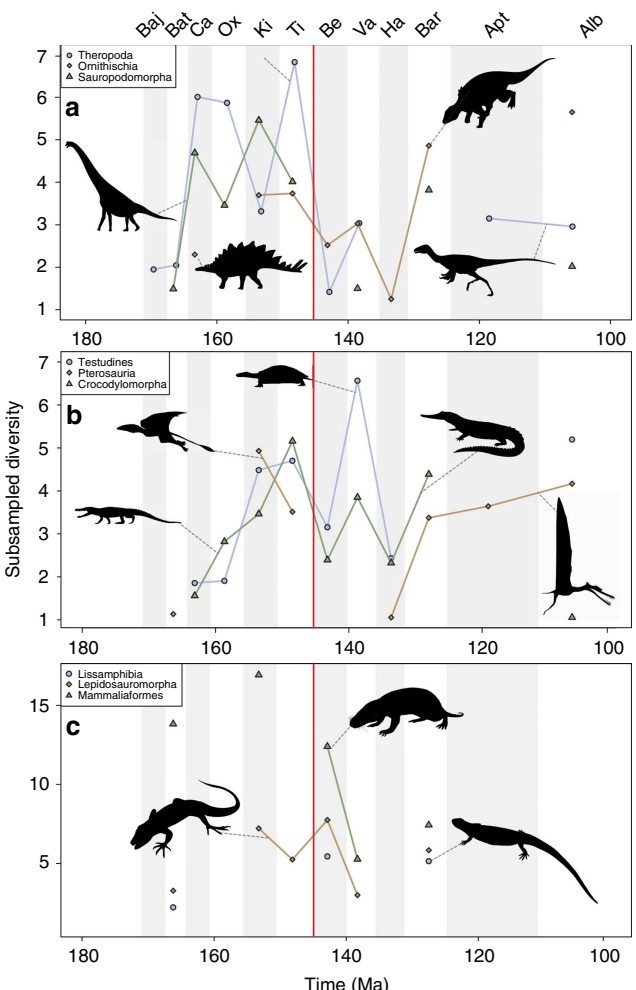

**Figure 2 | Global non-marine Late Jurassic to Early Cretaceous subsampled diversity.** SQS diversity for (**a**) major dinosaur groups; (**b**) testudines, crocodylomorphs and pterosaurs; and (**c**) mammals, lepidosaurs and lissamphibians. Shaded areas represent stage boundaries. Where gaps in the curve exist this is due to poor sampling and failure to adequately recover a subsampling diversity estimate. Silhouettes from Phylopic courtesy of Michael Keesey, Grad McFeeters, Scott Hartman, Mark Witton, Ville Veikko Sinkkonen and Hanyong Pu (see http://phylopic.org/ for additional license information).

subsampled results. Sauropods show the greatest evidence of decline across the J/K boundary, with a moderate decline in Theropoda, whereas ornithischian diversity remained stable (Supplementary Fig. 4). The results here contrast with those recently obtained using the novel TRiPS method[37], which found that each of the three major dinosaur clades did not suffer a diversity loss over the J/K boundary when simultaneously calculating both sampling rate and richness. The relative performance of TRiPS to other diversity estimation methods is beyond the scope of this paper, but is a factor that requires future investigation.

Non-marine crocodyliforms still document a loss of more than 50% of diversity[18] at the stage-level (Fig. 2b), but the magnitude of this decline could have been much greater based on our bootstrapped estimates (Supplementary Fig. 5). This drop in diversity was followed by the subsequent diversification of major non-marine clades such as Notosuchia and Eusuchia in the Hauterivian–Barremian[24]. Pterosaurs are too poorly sampled at the stage level to reveal an SQS signal across the J/K boundary,

but diversity in the Hauterivian was around 20% of Kimmeridgian levels, documenting low diversity subsequent to the extinction of non-pterodactyloid faunas[21,38]. However, a component of this low diversity signal is possibly due to anomalously low within-bin sampling of the Hauterivian, and pterosaurs steadily increased in diversity through the remainder of the Early Cretaceous (Fig. 2b), documenting the diversification of ornithocheiroid pterodactyloids[38]. This overall pattern is quite similar to that obtained from the PDE, with pterosaurs showing a major diversity drop across the J/K transition overall, a slight increase across the J/K boundary that the SQS results do not contradict, and then a substantial recovery in the Hauterivian. Non-marine turtles declined by 33% of diversity through the J/K boundary (Fig. 2b), in contrast to results obtained at a coarser resolution by ourselves and by a recent study[19], which found steadily increasing diversity. Some of this discrepancy with the study of Nicholson et al.[19] might also be due to differences in our treatment of coastal and freshwater taxa, which we regard as non-marine as opposed to fully marine (that is, exclusively chelonioids (sea turtles)). Mammals suffered an overall loss in global subsampled diversity of 69% from the Kimmeridgian to the Valanginian (Fig. 2c), similar to that recently recovered from subsampled and residual diversity estimates[32], but we do not recover the earliest Cretaceous rise in diversity reported by Newham et al.[32]; this is likely due to either our finer division of time bins (see above), or modified version of SQS (Supplementary Materials). Distinct from this broader pattern of decline, lepidosauromorphs greatly increased in diversity (48%) across the J/K boundary (Fig. 2c), reflecting the diversification of major extant squamate clades, including Lacertoidea, Scincoidea and Iguania, in the earliest Cretaceous[25,39]. Lissamphibian diversity is consistently low but discontinuously resolved through the J/K boundary (Fig. 2c), reflecting overall poor Mesozoic sampling of this group.

In the marine realm, stage-based crocodyliform SQS diversity decreased by around 50% (Fig. 3), reflecting the ongoing decline of Thalattosuchia before their extinction in the Early Cretaceous[29,35,40]. The magnitude of this decline is slightly greater than that reported by some previous studies[18,41], most likely corresponding to our usage of a finer resolution timescale, which reduces the lumping of non-contemporaneous taxa. Indeed, our subsampled results suggest that the level of diversity decline could have been much greater (see also the PDE results in Supplementary Data 3) and perhaps even initiated in the Kimmeridgian–Tithonian. Ichthyosaurs and sauropterygians are too poorly sampled in the earliest Cretaceous, but when an SQS signal emerges in the Early Cretaceous (Hauterivian and Valanginian, respectively), diversity is consistently <50% of Late Jurassic levels (Fig. 3), similar to previous estimates using residual diversity[15,22]. The conclusions drawn from these marine and non-marine results do not change markedly if we vary the quorum, although the magnitude of diversity decline increases with higher quorum levels (Supplementary Data 4), and bootstrapping does not alter the patterns across the J/K transition (Supplementary Fig. 6). Moreover, based on PDEs, both sauropterygians and ichthyosaurs show evidence for a notable decline in diversity across the J/K boundary, which continued into the Hauterivian for both groups (Supplementary Fig. 7).

Notable differences between some of the results from our stage-level and 10 million year time bin analyses emphasise the effect that choice of time binning can have on our understanding of the magnitude of the J/K boundary extinction. In particular, whereas 10 million year bins might give a 'fairer' method of grouping data, their relative coarseness means that some key aspects of palaeobiodiversity patterns are obscured

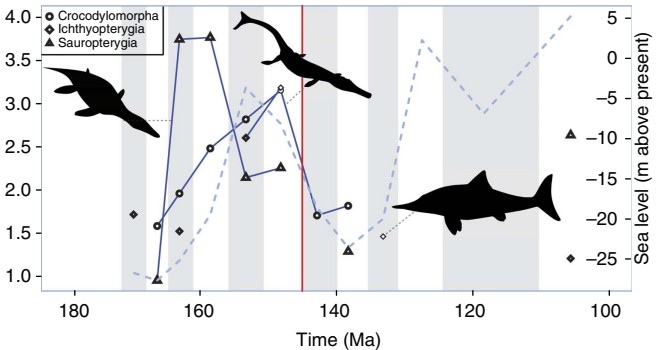

**Figure 3 | Global marine SQS diversity for major Late Jurassic to Early Cretaceous pelagic clades.** Sea level curve from Miller et al.[53]. Silhouettes from Phylopic courtesy of Gareth Monger and Michael Keesey (see http://phylopic.org/ for additional license information).

(Supplementary Methods). Previous studies of tetrapods have failed to find any correlation between time bin length and raw diversity[15,18,20,22], suggesting that stage-level bins are appropriate for diversity studies (that is, diversity does not systematically increase with bin length, leading to artificial overestimation). Furthermore, the time bin lengths for the Kimmeridgian, Tithonian, Berriasian and Valanginian are quite similar (3.8–5.3 million years), and therefore interpreting stage-level patterns through this interval should be sufficient to recover fair estimates of subsampled diversity.

**Patterns of global extinction and origination.** The global patterns of diversity reported above are reflected in rates of extinction and origination. Crocodyliforms (marine and non-marine), ichthyosaurs, sauropterygians, sauropods, theropods, pterosaurs and turtles all experienced greatly elevated extinction rates at the end of the Tithonian, recorded in both boundary-crosser and three-timer estimates (Supplementary Data 5). However, in some groups (for example, ichthyosaurs) it is still possible that an extremely poor earliest Cretaceous fossil record influences these results. In most groups, the earliest Cretaceous record is too poor to gauge accurate estimates of extinction. However, marine crocodyliforms, theropods and pterosaurs sustained high extinction rates into the Berriasian, which coincides with the greatest diversity losses recorded among all tetrapod groups during our study interval. The highest end-Jurassic extinction rates were in sauropods, at more than six times Jurassic turnover rates, and twice those in the 'middle' Cretaceous. In all other groups, the end-Jurassic was singularly the most intense period of extinction of all intervals throughout the Jurassic and Cretaceous (except for the end-Cretaceous mass extinction, which was not included in our analyses).

Origination rates in almost all groups were severely depressed in the earliest Cretaceous which, when combined with the high latest Jurassic extinction rates, explains the consistently low diversity recorded in most clades. Despite elevated origination rates in some groups during the Oxfordian (for example, thalattosuchians) and Kimmeridgian (for example, non-marine crocodyliforms, all dinosaurs and pterosaurs), at between 2–10 times the rates of other intervals during the Jurassic, these do not appear to have conferred any survivorship advantage through the J/K boundary. However, relatively high origination rates in ichthyosaurs and sauropterygians during the Kimmeridgian–Tithonian, representing the radiations of Platypterygiinae[30,31] and Xenopsaria[17], respectively, might have been responsible for their moderately high apparent survival rates through the J/K boundary based on 'ghost lineages'[17,30], following a

contemporaneous diversity crash in cryptoclidid plesiosaurs[17]. This high extinction pattern for ichthyosaurs is distinct from that recovered by Fischer et al.[30], who noted either suppressed extinction rates or little to no deviation from background rates across the J/K boundary based on a boundary-crosser estimate. Fischer et al.[30,42] argued that the earliest Cretaceous was a period of quiescence for ichthyosaurs, representing a phase of moderate diversity but no speciation. However, a more likely explanation, supported by our results that account for the 'Signor–Lipps' effect (that is, the artificial prolonging of extinction events), is that there were high extinction rates in ichthyosaurs in the latest Jurassic that led to their apparently low diversity throughout the earliest Cretaceous (when we are able to recover a signal), broadly consistent with other marine tetrapod groups.

**Regional patterns of diversity across the J/K boundary**. It is important to determine whether or not these apparently global patterns are the product of grouping together disparate regional palaeocontinental level signals. The most comprehensive record from the Jurassic–Cretaceous interval comes from Europe (Fig. 4a,b). Here non-marine crocodyliforms experienced a 'double-dip' diversity decline, with troughs in the Berriasian (37% loss) and Hauterivian (40% loss), before a major Barremian diversification[24]. This pattern is similar to that in turtles, which document a decline of 38% over the J/K boundary, with a recovery to their highest Mesozoic levels in the Valanginian, before a second, more severe diversity decline (69%) in the Hauterivian. Lepidosaurs showed increasing diversity through the J/K boundary and, although lissamphibian diversity patterns remain obscured, they were twice as diverse during the Barremian as the Tithonian. European mammal diversity is poorly resolved at the stage level in the latest Jurassic, although this group suffered a loss of 58% of diversity from the Berriasian to the Valanginian. Ornithischian diversity increased through the J/K boundary, but steadily declined from the Berriasian to the Hauterivian (51% loss). Both sauropods and pterosaurs were in decline in Europe before the J/K boundary and, although their earliest Cretaceous dynamics are unknown, their diversity is consistently lower in the Hauterivian–Barremian than in the latest Jurassic. Theropods lost 76% of their diversity from the Kimmeridgian to Berriasian, but recovered rapidly in the Valanginian. Much of this European signal is reflected in the overall 'global' pattern of decline recovered in all non-marine tetrapod groups. Considering the poor sampling of North America, Asia and Gondwana during the earliest Cretaceous (see below), this indicates that most of this apparently global decline is the result of a regional signal focussed in Europe. Furthermore, instead of a globally synchronous event across the J/K boundary as implied by our global analyses, the European pattern suggests that the tempo of decline was staggered, with diversity decreasing in a cascading manner across the J/K transition. Therefore, global patterns of tetrapod diversity are probably poor indicators of regional-level dynamics, and care should be taken to distinguish between these signals.

In the Cretaceous of North America, diversity patterns can only be reconstructed for theropods and ornithischians in the late Early Cretaceous, which were both more diverse than their Late Jurassic counterparts, with all other groups too poorly sampled to retrieve a signal. In the non-marine record of North America, there are similar problems to Gondwana (see below) with distinguishing between false absences (that is, a sampling failure) and true absences (that is, a genuine lack of fossil occurrences and diversity) in the Early Cretaceous, with implications for the spatial structure of terrestrial diversity over the J/K boundary[18,19]. The North American non-marine fossil record is temporally

discontinuous, but within a continuous macrostratigraphic sequence. This implies that sedimentary rock is available for sampling even during times when fossil record sampling and diversity are low or nil[43,44] (Fig. 4c,d), which suggests one of two possibilities: either (1) the environments in which earliest Cretaceous tetrapods lived or were fossilised are not preserved in the available rock record; or (2) the lack of tetrapod fossils in the earliest Cretaceous represents genuine absence, following a J/K boundary extinction.

In Asia, the record of tetrapod diversity dynamics through the J/K boundary is patchy and discontinuous, with a Late Jurassic non-marine record composed primarily of crocodyliforms and lissamphibians possibly being replaced by an Early Cretaceous one comprising lepidosaurs and choristoderes. Mammals, non-avian theropods and birds appear to have radiated explosively during the Aptian in Asia, documented by the exceptionally well-preserved Jehol Biota[45], whereas sauropod and ornithischian diversity remained comparatively low. This high diversity in small-bodied forms is undoubtedly influenced by the Lagerstätten effect (that is, episodes of greatly enhanced fossil record preservation). However, the high diversity of this fauna remains in spite of our application of a fair subsampling protocol, and therefore we infer that this pattern is a genuine biological signal.

In Gondwana, there is almost no information about diversity in the earliest Cretaceous (Supplementary Data 3). The only group for which we can recover a signal in the Berriasian of Africa is Theropoda, which documents a regional diversity decline of around 50% across the J/K boundary. Fossils from the earliest Cretaceous of Africa are almost entirely absent, virtually exclusively known from dinosaurian faunas of the Berriasian–Valanginian Kirkwood Formation[46] of South Africa, and a Berriasian vertebrate microsite in Morocco[47]. The Moroccan site at Anoual includes a diverse assemblage of mammals, lissamphibians, lepidosaurs and turtles, as well as numerous indeterminate dinosaurs, the majority of which represent small maniraptoran theropods[47]. However, many of these data are not added to total diversity estimates because they are single-publication occurrences from one large collection (Supplementary Methods and Alroy[34]), and therefore the tetrapod diversity of the earliest Cretaceous of Africa is underestimated[46] in our results. In South America, we can only detect moderately low diversity for turtles, pterosaurs and sauropods in the Late Jurassic, but a signal does not emerge again until the late Early Cretaceous. This overall pattern is similar to that of North America, with a well-sampled Late Jurassic fauna discontinuously sampled through the J/K boundary, with no signal emerging until the Barremian. The main issue for Gondwana is whether the overall lack of signal results from either sampling failure or genuinely low diversity, a problem exacerbated by the lack of a comprehensive geological record throughout this time.

Our understanding of marine tetrapod dynamics across the J/K boundary is dominated by the South American and European fossil records, with the marine records in North America, Africa and Asia too discontinuous to document changes. In South America we see a small decline in thalattosuchian diversity (10%), coupled with an apparent loss of all ichthyosaur and sauropter-ygian taxa, but we note that some poorly dated taxa were excluded from our analyses as they cannot be constrained to any single time bin. For intervals in which sampling is too poor to produce a subsampled diversity signal, we report a result of non-applicable (that is, a gap in our knowledge). We acknowledge that even in these intervals there are often specimens present, but that these will just be singleton occurrences, or not taxonomically identifiable to the genus level. In Europe we observe the greatest loss in diversity, similar to the non-marine realm, with thalattosuchians showing a major decline (52%) alongside

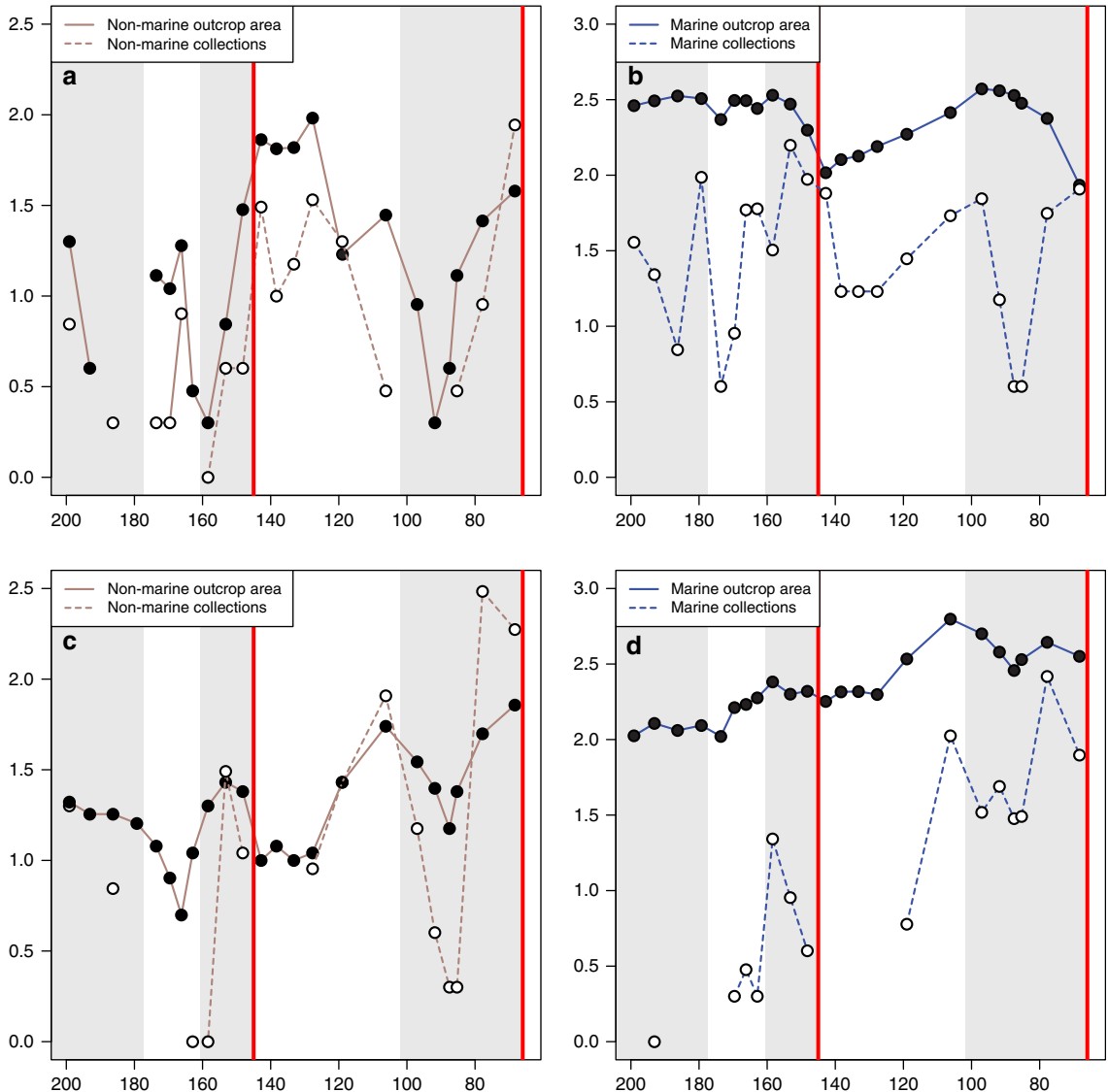

**Figure 4 | Regional outcrop area and collection counts.** For the (**a**) non-marine record of Europe; (**b**) marine record of Europe (**c**) non-marine record of North America; and (**d**) marine record of North America. European outcrop area from Smith and McGowan[49], and North American outcrop area from Peters and Heim[43]. Note the discontinuity between the availability of the rock record and the number of collections in the earliest Cretaceous of North America.

sauropterygians (48% loss) from the Tithonian to Valanginian. Ichthyosaurs are too poorly sampled in the earliest Cretaceous, but their Hauterivian diversity is 40% that of Tithonian levels, a signal that we are able to recover with SQS at a moderate quorum (0.4), despite a relatively low sample size. This overall lack of signal in the Berriasian marine record can be attributed to a sampling failure (Good's $u = 0$). However, sampling in the Valanginian is globally better than the latest Jurassic (Kimmeridgian–Tithonian), indicating that the overall pattern of an apparently global decline in the marine realm reflects a regional biological signal from South America and Europe.

**The impact of sampling and redundancy on diversity.** The relationship between raw empirical diversity and estimates of global sampling based on tetrapod-bearing collections and formation counts for the non-marine (TBC and TBF, respectively) and marine (MBC and MBF, respectively) realms (Supplementary Data 6 and the Supplementary Methods) is almost consistently positive and strong for each taxonomic

group analysed, even after correcting for false positives (that is, the false-discovery rate). This pattern is found in both the marine (for example, for ichthyosaurs: Akaike Information Criterion (AICc) weight $= 0.92$; Pearson's $r = 0.877$; adjusted $P = 0.04$) and terrestrial (for example, for mammaliaforms: AICc weight $= 0.836$; Pearson's $r = 0.751$; adjusted $P = 0.018$) realms. In every group for which a strong relationship between empirical taxonomic diversity and sampling is not found, there is also no strong relationship recovered for any of the other extrinsic parameters. Therefore, the correlations between raw diversity of J/K tetrapods and sampling either reflects the effect of redundancy or demonstrates that sampling controls observed diversity[10,13]. The possible issue of redundancy arises from the non-independence of a sampling proxy with diversity[10,11,13], which should be alleviated by our use of a higher-level proxy such as tetrapod-bearing formations[32,48], as this captures more information about potential opportunities to sample that might be missed by a lower level proxy (Supplementary Methods). As such, redundancy appears to be the less likely explanation for these correlations.

                                                                    

We also assessed the impact of rock outcrop area as a non-redundant proxy for geological sampling on the shape of both empirical and subsampled diversity curves (Supplementary Data 7). This was only possible on a regional level (that is, for North America and western Europe), where readily available estimates of rock outcrop area exist[43,49]. In Europe, raw, summed, non-marine tetrapod taxonomic richness is strongly correlated with non-marine western European rock outcrop area (Spearman's rho $= 0.671$, adjusted $P = 0.034$). This pattern is distinct from the North American record, in which raw marine tetrapod richness does not correlate with non-marine outcrop area.

In contrast, when we compare global subsampled diversity with these global sampling metrics, for almost every group no statistical relationship exists for either collections or formations (Supplementary Data 7). The only exception to this pattern is theropod diversity, which retains a strong, positive correlation with global counts of non-marine collections even after subsampling (Pearson's $r = 0.79$, adjusted $P = 0.044$). However, rather than interpreting this as an instance of sampling controlling the shape of subsampled theropod diversity, we consider it more likely to reflect the fact that theropods are so well sampled globally that they closely mirror the global tetrapod record. As we find that subsampled diversity, as a more accurate estimate of true diversity[5,34], is independent of higher taxonomic-level sampling metrics (that is, MBFs/TBFs and MBCs/TBCs) at a global level (Supplementary Data 6), this means that we can reject the hypothesis that 'true' diversity controls sampling, and so does not control our sampling metrics.

Furthermore, for each taxonomic group in which there is a sufficiently continuous subsampled European record through the J/K boundary, there is no relationship between either European marine or non-marine outcrop area and subsampled diversity (Supplementary Data 7). European outcrop area is also independent of any other regional sampling metric, which probably reflects Europe having a more intensive sampling history compared with the rest of the world. Furthermore, an estimate of European fossil record coverage based on Good's $u$ (Supplementary Methods) is only weakly and nonsignificantly negatively correlated with western European marine (Pearson's $r = -0.429$, adjusted $P = 0.625$) and non-marine (Pearson's $r = -0.348$, adjusted $P = 0.267$) outcrop area (Supplementary Data 6). Similarly to Europe, no individual North American tetrapod group for which there is a sufficiently continuous record (only Ornithischia and Theropoda) exhibits a strong relationship between subsampled diversity and non-marine outcrop area. Good's $u$ is also not correlated with either marine or non-marine outcrop area in North America, suggesting that increasing regional outcrop area (that is, geological sampling) has little to no effect on the overall evenness and structure of tetrapod sampling for reconstructing diversity, irrespective of whether this means there are more opportunities to sample (collections) or not. Marine outcrop area in Europe shows no significant relationships with any of our sampling metrics, or with raw or subsampled diversity, which is broadly congruent with our global level analyses. Given that our regional rock record metrics and subsampled diversity estimates are shown not to be the product of 'redundancy'[13] (a similar conclusion to that reached by Upchurch et al.[16] for North America and Europe), this further implies that our potentially redundant proxies (that is, formation and collection counts) are capturing a genuine regional sampling signal. This provides support for 'correcting' diversity curves by choosing a 'higher-level' proxy that accounts for any potential redundancy. Such a conclusion is supported by studies that show a close short-term relationship between diversity curves produced using both SQS and residual estimates using sampling proxies[50].

The results outlined above suggest that on a global level, both geological and anthropogenic sampling appear to control raw taxonomic diversity, but this is alleviated when subsampling is applied, as observed from the switch from almost universally significant positive correlations to no correlations (Supplementary Data 7). Although we urge caution in the interpretation of nonsignificant results as evidence for no relationship, this shift in correlation strength occurs in almost every taxonomic group, independently of their sampling histories and overall diversity patterns. Therefore, although SQS was designed to account for collection-based sampling issues[5,34], and problems relating to geological sampling biases were implicitly ignored (despite their wide documentation[43,50,51]), this method seems to also alleviate issues pertaining to geological sampling variation. These results collectively suggest that SQS is an adequate method to account for fossil record bias, as opposed to scrambling a common underlying signal influencing both sampling and diversity[52]. This utility of SQS most likely occurs because the relationship between regional collection counts and outcrop area is almost consistently strongly positively correlated in the non-marine and marine realms (Supplementary Data 7); the exception to this is the marine record of Europe, in which outcrop area appears to be independent of raw and subsampled tetrapod diversity.

**Testing the common cause hypothesis**. Global sea level is uncorrelated with global non-marine tetrapod-bearing formations (Pearson's $r = 0.034$, $P = 0.92$) and collections (Pearson's $r = 0.301$, $P = 0.368$), and marine tetrapod-bearing formations (Pearson's $r = 0.195$, $P = 0.566$) and collections (Pearson's $r = 0.222$, $P = 0.512$), and therefore we can reject the hypothesis that sea level acts as a common factor driving both global sampling and diversity of marine and non-marine tetrapods. In addition, our results imply that regional rock records and estimates of subsampled regional diversity are not connected by a 'common cause' factor such as sea level[12], with the possible exception of the marine realm in North America.

In North America, both marine and non-marine outcrop area are strongly correlated with fluctuations in eustatic sea level[53] (Pearson's $r = 0.702$, adjusted $P = 0.01$ and Pearson's $r = 0.63$, adjusted $P = 0.04$, respectively). However, the marine tetrapod record is too patchy to detect any statistical relationship between outcrop area and subsampled diversity in this region. This implies that sea level exerts a strong control on the geological record of North America, but we cannot determine whether or not this relationship influences regional subsampled diversity estimates. Therefore, we cannot discount a 'common cause' relationship influencing marine diversity estimates in North America because of the discontinuous nature of the fossil record[9]. Unlike the latter region, sea level is not related to European marine or non-marine outcrop area, similar to the results of Dunhill et al.[11]. However, we find no relationship between western European non-marine outcrop area and the number of tetrapod-bearing formations (Pearson's $r = 0.328$, adjusted $P = 0.256$), distinct from the relationship recovered by Dunhill et al.[11], who examined an exclusively British fossil record. Furthermore, we find no relationship between marine outcrop area and tetrapod-bearing marine collections or formations for western Europe, suggesting that the signal recovered by Dunhill et al.[11] is strongly localized to the unique collecting and tectonic histories of Britain, where sampling has been focused on historical mining and collections from ephemerally exposed localities along coastlines.

**Extrinsic drivers of Jurassic/Cretaceous tetrapod diversity.** Eustatic sea level is shown to be the principal mechanism controlling the 'true' Jurassic–Cretaceous diversity of tetrapods,

**Table 1 | Selected results of model-fitting procedure.**

| Group | Parameter | AICc | | Spearman's rank | | Pearson's PMCC | |
|---|---|---|---|---|---|---|---|
| | | Likelihood | Weight | Rho | Adjusted *P* value | *r* | Adjusted *P* value |
| Crocodyliformes (marine) | Palaeotemp. | 22.741 | 0.237 | − 0.524 | 0.634 | − 0.522 | 0.678 |
| Crocodyliformes (non-marine) | Sea level | 26.285 | 0.969 | 0.750 | 0.175 | 0.846 | 0.028 |
| Lissamphibia | Palaeotemp. | 38.260 | 0.796 | 0.700 | 0.301 | 0.742 | 0.154 |
| Mammaliaformes | Sea level | 51.394 | 0.931 | − 0.450 | 0.537 | − 0.666 | 0.301 |
| Ornithischia | Sea level | 60.106 | 0.391 | 0.200 | 0.681 | 0.047 | 0.898 |
| Pterosauria | Sea level | 33.261 | 0.872 | 0.714 | 0.406 | 0.647 | 0.581 |
| Sauropodomorpha | Sea level | 41.191 | 0.501 | 0.310 | 0.810 | 0.457 | 0.564 |
| Sauropterygia | Sea level | 41.820 | 0.409 | 0.055 | 0.906 | 0.065 | 0.985 |
| Testudines | Palaeotemp. | 50.648 | 0.258 | 0.343 | 0.880 | 0.462 | 0.891 |
| Theropoda | Sea level | 72.931 | 0.534 | − 0.018 | 0.968 | 0.037 | 0.954 |

For complete results for both subsampled and raw taxonomic diversity, see Supplementary Information 7. Data for sea level from Miller *et al.*[53], and for palaeotemperature (Palaeotemp.) from the $\delta^{18}$O proxy from Prokoph *et al.*[54].

being strongly positively correlated with subsampled diversity of lepidosaurs (AICc weight = 0.727), mammals (AICc weight = 0.931), ornithischians (AICc weight = 0.391), theropods (AICc weight = 0.534), sauropods (AICc weight = 0.501), pterosaurs (AICc weight = 0.872), sauropterygians (AICc weight = 0.409) and non-marine crocodyliforms (AICc weight = 0.969) (Table 1). The relationship between sea level and dinosaur diversity differs from that recovered by Butler *et al.*[12], who found no correlation between detrended fluctuations in dinosaur diversity and sea level, which can most likely be explained by our contrasting approaches to reconstructing diversity (that is, SQS here versus residuals in Butler *et al.*[12]). However, our finding is more congruent with other studies that have documented sea level as the principal controlling factor on Phanerozoic diversity (for example, ref. 52). In the case of marine crocodyliforms, their subsampled diversity was driven by a combination of factors, including sea level, nutrient cycling and eustacy-influenced redox shifts[35]. In contrast, lissamphibian diversity shows a strong positive correlation (AICc weight = 0.796) with palaeotemperature[54], with weaker support in non-marine turtles (AICc weight = 0.258)[19], suggesting that these semi-aquatic groups are more sensitive to palaeoclimatic shifts than to changes in eustatic sea level. This relationship between palaeotemperature and lissamphibian and turtle diversity could be due to a late Tithonian 'cold snap'[55], followed by a global temperature increase during the Berriasian[56], which are possible candidates for causing these groups to decline and then radiate during the earliest Cretaceous[19]. Ichthyosaur diversity is negatively correlated with global subsampled marine invertebrate diversity (AICc weight = 0.42), suggesting that the global richness of the former is tied to broader patterns influencing diversity in the marine realm rather than to a possible food source. For poorly sampled groups, such as birds, choristoderes and marine turtles, we were unable to resolve the controls on their diversity patterns. Where we recover relatively lower AICc weights, this indicates that additional parameters that we did not analyse here, such as post-extinction opportunism or competitive displacement[18], or passive aspects of trait evolution[57], might also have played a significant role in affecting global diversity patterns for certain groups. Alternatively, diversity in these groups might be driven by a combination of factors, rather than any single underlying diversity regulator.

The relationship between sea level and tetrapod diversity has previously been examined in most detail for dinosaurs[12,20,58] and crocodyliforms[18,41], and our results lend strong support to these earlier studies, reinforcing the view that changes in sea level control the architecture of near-shore ecosystems. Support for this conclusion in a range of groups with vastly different ecologies, from pelagic open ocean swimmers and volant taxa, to small- and large-bodied terrestrial groups, suggests that sea level influences these groups in a variety of ways. A relationship between sea level and terrestrial diversity can best be explained by rising sea levels leading to greater division of landmasses through creation of marine barriers. This alters the spatial distribution of near-shore habitats and affects the species–area relationship, which can lead to elevated extinctions. Such fragmentation can also be a potential driver of biological and reproductive isolation and allopatric speciation, the combination of which we would expect to see manifest in the diversity signal. However, evidence for these potential relationships between sea level, terrestrial diversity and sampling has previously remained elusive[12]. As we find evidence for a positive correlation between sea level and diversity in multiple terrestrial clades, this suggests that allopatric speciation has outweighed the species–area effect for non-marine tetrapods during our study interval. Furthermore, the diversity of fully marine taxa was more probably affected by the opening and closure of marine dispersal corridors, whereas that of terrestrial and coastal taxa was more probably dependent on the availability of habitable ecosystems, including the extent of continental shelf area. However, the global extent of this relationship between sea level and diversity is difficult to discern, and confounded by issues of linking a global parameter like eustatic sea level with spatially heterogeneous diversity patterns and sampling regimes. Irrespective of this, there is strong evidence that a eustatic lowstand across the J/K boundary impacted on global marine and non-marine faunas, a phenomenon that is most clearly marked in the European data[33]. These changes in sea level can be attributed to a first-order transgressive–regressive cycle driven by the ongoing fragmentation of Pangaea, and geothermal uplift at mid-oceanic ridges[53,59], and has previously been proposed to have driven regional extinctions across the J/K boundary[33,60]. Thus, we cannot rule out that tectonic reconfiguration was the driver of both continental breakup and eustatic changes, and therefore ultimately played a key role in determining tetrapod diversity patterns during the J/K transition.

## Discussion
We have demonstrated that both marine and non-marine tetrapod faunas show evidence for a global ecological and taxonomic reorganization across the J/K boundary. Whereas the diversity of groups such as pterosaurs and sauropods began to fall before the J/K boundary, and that of others such as mammals and ornithischians decreased subsequently in the earliest Cretaceous, the majority of clades document their greatest Jurassic–Cretaceous decline through the boundary itself. The magnitude

of this drop in diversity ranges from around 33% for ornithischians to 75–80% loss for theropods and pterosaurs. This is coupled with elevated extinction rates, almost at the level of mass extinction, and strongly depressed origination rates throughout the earliest Cretaceous that are sufficiently distinct from rates throughout much of the rest of the Jurassic and Cretaceous to warrant future investigation. Together, this is strong evidence for several pulses of extinction and radiation, culminating in a 'wave' of ecological turnover through the J/K boundary. Ultimately, this could be related to the radiation of several important clades during the earliest Cretaceous, including birds, lissamphibians and several groups of semi-aquatic turtles.

Although we have identified eustatic sea level as the principle driver behind these patterns, the wider implications within a total ecosystem context need to be considered. The J/K boundary saw a major revolution in marine microorganism communities that has been attributed to increasing global aridity and continental weathering[61,62], culminating in increasingly oligotrophic conditions in the marine realm[63,64]. It is likely that such environmental changes were primarily related to the sea-level regression that occurred across the J/K boundary[33], which together impacted on global ecosystems. In addition, there is a range of singular but potentially more catastrophic events that will require factoring in to future investigations of the faunal turnover during the J/K transition. These include the Morokweng bolide impact in South Africa at the J/K boundary[65], as well as numerous episodes of Early Cretaceous flood volcanism[6,66], including the emergence of the Ontong Java Plateau, which was potentially a more marked volcanic event than that linked to the end-Cretaceous mass extinction, and might have played an important role in the evolution of tetrapods throughout the Early Cretaceous. Combining our understanding of small-scale microorganism communities and evidence for large-scale catastrophic events with the patterns that we have recovered here should provide a more detailed appreciation of the complexity of the Jurassic/Cretaceous interval.

## Methods

**Tetrapod occurrences data set.** A comprehensive overview of materials and our analytical protocol can be found in the Supplementary Methods. Our data set is based on a new fossil occurrence compilation[67] (Fig. 1 and Supplementary Fig. 2) that spans the entirety of the Jurassic to Cretaceous (201–66 Myr ago). This comprises a near-comprehensive record of published fossil occurrences of tetrapods within the *Paleobiology Database* (PaleoDB; http://www.paleobiodb.org/), following extensive work to ensure that occurrences and taxonomic opinions reflect our current published knowledge[67] (Supplementary Data 1). Fossil occurrences that could be assigned to genera were downloaded from the PaleoDB (accessed 31 May 2015), including those with qualifiers such as 'aff.' and 'cf.', and totalled 12,476 occurrences from 10,985 collections. We selected to use genera to allow inclusion of specifically indeterminate occurrences within our data set. These data were then subdivided into time bins using two different binning strategies: (1) at the stage level, comprising 7,312 occurrences from 6,316 collections, representing 1,275 genera drawn from 2,313 published references; and (2) at the 10 Myr level, comprising 10,874 occurrences from 9,454 collections, representing 1,954 genera drawn from a total of 3,774 published references (Supplementary Table 3). Note that this means that our data set can exclude even well-known specimens or taxa if they are poorly constrained temporally. Our geological time binning scheme is based on the Standard European Stages and absolute dates provided by Gradstein *et al.*[68]. The reason for our combined time binning scheme is that the former provides a finer scale resolution for investigating changes in diversity, whereas the latter ensures that time bins sample occurrence data at even time intervals (NB Jurassic and Cretaceous time bins are of uneven length, ranging from around 2–13 million years). Occurrences were divided into marine and non-marine partitions, with marine taxa representing only those which were fully pelagic. Semi-aquatic and coastal taxa were treated as non-marine in all cases. Where time bins did not contain any occurrence data, these were treated as non-applicable data, rather than 0 data. Taxonomic groups are based on major clades that either passed through the J/K boundary or radiated in the Early Cretaceous (Supplementary Table 1). Each taxonomic subgroup was further sub-divided into approximately contiguous palaeocontinental regions: Africa; Asia; Europe; South America; and North America (Supplementary Data 1). Sampling is too poor to analyse patterns in Antarctica, Australasia or Indo-Madagascar,

although these regions were included in our global analyses. Each fossil occurrence has an associated stratigraphic range based on the temporal duration of its parent collection, which in turn is based on the geological strata from which that collection is sampled. We used this to assign individual minimum and maximum ages to each occurrence. Only occurrences that had their entire stratigraphic range contained within a single time bin were included. This approach avoids the over-counting of single occurrences in multiple time bins and the spurious inclusion of taxa with high uncertainty in their temporal durations.

**Subsampling protocol.** SQS standardizes in-bin taxonomic occurrence samples based on an estimate of coverage to determine the relative magnitude of taxonomic biodiversity trends[34]. In this subsampling approach, each taxon within a time bin is treated as a 'shareholder', whose 'share' is its relative occurrence frequency[34]. Taxa are randomly drawn from lists compiled for each bin, and when a summed proportion of these taxa and their associated 'shares' reaches a certain threshold, or 'quorum', subsampling ceases and the number of subsampled taxa is summed. Coverage is defined as the proportion of the frequency distribution of taxa within a sample, and estimated by using randomized subsampling to calculate the mean value of Good's $u$ (refs 5,34,69). Note that as the current application of SQS only returns mean values, we are unable to assess whether or not the resulting changes in diversity are statistically significant, as has been the case in all previous applications of this method. Therefore, we used a novel bootstrapping method to assess the sensitivity of this subsampling approach (Supplementary Data 8). SQS was applied to each occurrence data set for our higher taxonomic groups to estimate global subsampled diversity. This was conducted for both of our binning strategies (see above), and implemented using a Perl script provided by J. Alroy (version 4.3) (Supplementary Data 9). We executed 1,000 subsampling trials for each group, and report the mean diversity. For each sequential subsampling iteration, whenever a collection from a new publication was sampled from the occurrence list, subsequent collections were sampled until exactly three collections from that publication had been selected[34]. We set a baseline quorum of 0.4, and use the results from these as the basis for modelling our extrinsic parameters (see below). Full results, including those at different quorum levels and using a bootstrapping protocol, are reported in Supplementary Data 3 and 4.

**Extinction and origination rates.** Extinction and origination rates were calculated for the global occurrence data sets for each higher taxonomic group based on two different measures, 'Foote' rates (a boundary-crosser method), and three-timer rates (see Supplementary Methods and Supplementary Data 10 for further information).

**Sampling proxy data.** We sourced a range of sampling proxy data from the primary literature. Sampling proxies have been broadly utilized to capture some aspect of sampling, mostly regarding geological and anthropogenic factors, and used to 'correct' raw diversity curves for structural variations in sampling through time[10–13,15,16,21,22,32,50]. We calculated the number of tetrapod-bearing formations at the global level, as well as on a regional level for North America and Europe. Tetrapod-bearing formations are defined as any formally named geological formation that has ever yielded a published tetrapod body fossil occurrence, based on records compiled within the PaleoDB (Supplementary Data 1). These formation counts were divided into marine and non-marine partitions, based on whether or not marine and non-marine fossils occurred within them. Some marine formations were included in the non-marine tetrapod-bearing formations count because they have yielded some non-marine tetrapod fossils, and therefore represent opportunities to sample the latter. We also calculated the number of tetrapod-bearing collections, again at global and regional scales, as a metric for the intensity of anthropogenic sampling (that is, worker effort), and divided these into marine and non-marine partitions. For North American outcrop area, we used the COSUNA data set, which represents coverage of marine and non-marine geological units[43]. For western Europe, we used a proxy derived from an equal-grid sampling method of outcrop areas derived from geological maps[49]. Outcrop area represents a non-redundant proxy for the amount of sedimentary rock potentially available for sampling fossils. For a detailed discussion on sampling and the impact of megabiases on the fossil record, see the Supplementary Methods.

**Model-fitting procedure.** We extracted a range of environmental variables from the primary literature (Supplementary Data 1) to test whether extrinsic factors were the drivers of tetrapod diversity dynamics. These environmental proxies include the following: (1) eustatic sea level[53]; (2) palaeotemperature ($\delta^{18}$O) (ref. 54); (3) the global carbon ($\delta^{13}$C) cycle[54]; (4) the global sulphate ($\delta^{34}$S) cycle[54]; (5) the global strontium ($^{87}$Sr/$^{86}$Sr) cycle[52,54]; and (6) an estimate of global subsampled marine invertebrate biodiversity[52]. These environmental parameters were previously presented at the stage level, so were transformed into 10 Myr time bin data by taking the arithmetic mean of values for groups of data points that fall within the individual time bin intervals.

The residuals of each of these environmental parameters were calculated by using the arima() function, which uses maximum likelihood to fit a first-order autoregressive (AR(1)) model to each time series[70]. This method removes the influence of any long-term background trend (that is, a directed change in the

mean value of the total time series through time) within the data set, which can artificially inflate correlation coefficients[71], and also accounts for serial autocorrelation (that is, the correlation of a variable with itself through successive data points). A range of differencing techniques have been widely applied to correct time series when analysing fossil vertebrate data[10,12,15], and we use this method because the maximum likelihood fitting approach accounts for missing values in the time series (that is, not applicable), as opposed to treating them as zero data. The residuals of each time series were independently compared using linear regressions with each of our measures of diversity, using the lm() function. The relative fit of each variable was assessed using the sample-size-corrected AICc[72], by calculating the likelihood and weight for each environmental parameter as a way of assessing the probability of each one among the candidate set of models. In addition, we performed pairwise correlation tests between our diversity estimates and each environmental parameter using parametric (Pearson's product moment correlation coefficient ($r$)) and non-parametric (Spearman's rank ($\rho$)) methods. For each pairwise statistical hypothesis test, we report both the raw and adjusted $P$ values, the latter calculated using the p.adjust() function in R, and using the 'BH' model[73]. This procedure controls for the false-discovery test when performing multiple hypothesis tests with the same data set, which can inflate type 2 error (that is, to avoid falsely rejecting the null hypothesis). These adjustments were performed on 'families' of the data set, rather than on all correlation tests, as otherwise we run the risk of setting the pass rate for statistical significance too low. All analyses were carried out in R version 3.0.2 (ref. 74) unless specified otherwise.

**Data availability.** The raw data and analytical code are available in the Supplementary Files.

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

## Acknowledgements

We are grateful for the combined efforts of all those who have collected Jurassic-Cretaceous tetrapod data, and to those who have entered these data into the Paleobiology Database, especially J. Alroy, M.T. Carrano, R.B.J. Benson, and R.J. Butler. We also thank J. Alroy for providing the Perl script used to perform SQS analyses and J. Alroy, G.T. Lloyd, and D.B. Nicholson with assistance in applying the techniques used in this study. J.P.T. is funded by a NERC PhD studentship (EATAS G013 13). P.D.M.'s research was supported by an Alexander von Humboldt Research Fellowship and an Imperial College London Junior Research Fellowship. P. U. received funding via a Leverhulme Trust Grant (RPG-129). This is Paleobiology Database official publication number 267.

## Author contributions

All authors conceived and designed the research. J.P.T. and P.D.M. compiled the data. J.P.T. performed the analyses and prepared the figures. All authors discussed the results and contributed to writing the manuscript.

## Additional information

**Competing financial interests:** The authors declare no competing financial interests.

