## [Peer Review File · Nature Communications]

Reviewers' Comments:

Reviewer #1 (Remarks to the Author)

This is an enormously thorough piece of work, based on a great deal of effort in reviewing the literature and updating and correcting tetrapod data in the Paleobiology Database. The authors use generally accepted methods to process their data and to perform linear model fitting, and these approaches lead to their finding that most segments of their data can be explained best by sea level change - although some of the best passes are for clades where such an explanation is hard to comprehend, namely mammaliaforms, ornithischians, pterosaurs, sauropodomorphs, and theropods (Table 1).

However, I question the core claims of the paper for two reasons. First, because the results seem unexpected or hard to understand from fundamentals of ecology. For example, it is most unusual that diversity of marine crocodyliforms is controlled mostly by paleotemperature, whereas that of non-marine crocodyliforms is dependent on sea level. [I don't doubt the former, but very much query the latter, which goes against decades of detailed work extending back to Markwick, whose work, and all subsequent work is uncited.]

Second, the methods veer between reasonable and unreasonable. The authors are aware of recent literature, and acknowledge many of the criticisms of popular methods in controlling sampling, but then, after partly accepting concepts such as common cause, redundancy, and the failure of the residuals method, proceed to defend and continue to use such highly criticised methods. The core method used here is SQS, but this is justified only by repeating assertions by Alroy as a justification, not for fundamental reasons, and not addressing published concerns that it does not do what is assumed. Throughout the MS, it is noted how all these different approaches sometimes correlate, and often do not, and yet they are all accepted as somehow pointing to sampling inadequacies [If cumulative metrics, residuals, and SQS were effective metrics of sampling, they should more often correlate and render the same results than not.] In relegating the raw, empirical data to the Supplement (SI Fig. 1) and basing the entire analysis on SQS-modified results, the authors repeat the approach of recent published research, but it could all be a house of cards. As they note, their SQS-modified results differ substantially from previous work, often by members of the same research group, who used this, and other methods of data-manipulation ('sampling correction'). Maybe SQS is more sensitive than they admit to modest changes in the input data (which it should not be, as it is meant to be correcting for gaps variable sampling).

Some specific points

158-160: 'Previous studies of tetrapods have failed to find a correlation between time bin length and raw diversity, suggesting that stage level bins are appropriate for diversity studies.' Does it? The absence of a correlation simply means there is no correlation, and does not remove the possibility that short bins sample less than long bins. With such patchy sampling as in the J-K interval, you cannot tell that all sorts of confounding factors might not be occurring. Dividing diversity by bin length is still not excluded. Secondly, this statement ignores a point made earlier that a bin of any length might confound numerous non-contemporaneous faunas and lose rapid fluctuations in diversity. The similarity of bin length (line 162) is the more useful point.

274-277: Do you have any evidence that comparisons of raw diversity and raw collection counts or formation counts say anything about sampling? This information is largely relegated to the Supplementary Materials, but you should not refer to collection counts or formation counts as sampling metrics, as they likely are not. They are simply counts of collections and formations that mirror the counts of specimens and species as a result of a common history of data accumulation (redundancy).

285-301: Note that actual rock exposure (= availability of rock for collection) may be entirely unrelated to outcrop estimates, especially in heavily agricultural temperate areas like Europe.

302-304: 'On a global level, both geological and anthropogenic sampling appear to control raw taxonomic diversity, but this is alleviated when subsampling is applied, as observed from the switch from almost universally significant positive correlations to no correlations. You have provided no evidence for this. The fact that different collections, formations, and rock area metrics correlate variously with diversity perhaps suggests all those 'sampling metrics' are poor estimators of sampling failure. The best estimator is surely to identify the minimally known unknowns, in other words, Lazarus taxa and ghost ranges from phylogenies (see Simth 1994, 2007 etc). These point to bins with known missing lineage records - it's not great, but it's a start. All the other stuff is guesswork and assertion since all those signals are either mutually entwined (= redundant) or themselves probably more patchy and incomplete (e.g. rock area/ volume) than the raw diversity data.

304-307: Interesting - so SQS has additional properties, not only dealing with patchiness of collecting and recording into the PBDB, but also geological bias? This is weak.

314-315: 'The exception to this is the marine record of Europe, in which outcrop area appears to be independent of raw and subsampled diversity.' Interesting. So the record that geologists would likely a priori predict to be best of all - heavily sampled Europe, and European marine rocks? - turns out worst. Maybe the battery of tests is not doing what you think it is.

315-317: 'Furthermore, outcrop area is independent of any other regional sampling metric, which probably reflects Europe having a more intensive sampling history compared to the rest of the world.' Surprising result - surely this should make you think twice. It is more likely that the regional 'sampling' metrics are not sampling metrics at all, but redundant correlates of diversity, or meaningless variable numbers. If your assumptions and methods don't work in the best sampled portion of the record, I'd be worried about all the prior assumptions.

321-324: 'suggesting that increasing outcrop area (i.e., geological sampling) has little to no effect on the overall evenness and structure of tetrapod sampling for reconstructing diversity, irrespective of whether this means there are more opportunities to sample (collections) or not.' Does this then suggest that you might be best to work with the empirical signal rather than a data manipulation of uncertain effectiveness?

327-330: 'However, subsampled diversity, as a more accurate estimate of true diversity, is independent of higher taxonomic-level sampling metrics (i.e., MBFs/TBFs and MBCs/TBCs) at a global level (SI 6), meaning that we can reject the hypothesis that 'true' diversity controls sampling, and so does not control our sampling metrics.' Is there any evidence that SQS does all this? We only have Alroy's assertions of its properties, and nobody has provided fundamental evidence why. As you noted earlier, SQS was designed to deal only with human sampling bias, not geological bias, and surely geological patchiness is rampant throughout your J-K records (e.g. massive facies switch in Europe from marine to nonmarine at the J-K boundary), and it reduces the data to a much flattened-out version of the raw signal. SQS does not provide an absolute signal, merely a relative signal, although

that's fine for detecting rises and falls, but not really magnitudes of those rises and falls. SQS is likely sensitive to the exact bins chosen and the exact figures estimated - for example, Alroy's SQS estimates of global diversity are sensitive to the current state of the PBDB - add another tranche of data, and it alters massively. It shouldn't do this if it is truly accounting for sampling.

330-334: 'Nor are our regional rock record metrics and subsampled diversity estimates the product of 'redundancy', a similar conclusion to that reached by Upchurch et al. for North America and Europe, which further implies that our potentially redundant proxies (i.e., formation and collection counts) are capturing a genuine regional sampling signal.' That's a bold assertion. Just because you say it (and other people have said it) does not mean it is true - you have to provide evidence. If SQS is the wunder-metric, and SQS does not relate to any of the cumulative collections/ formations/ rock area metrics, then your statement is more likely false. [The following sentence is then redundant.]

351-356: ... or maybe Dunhill et al. found a strong correlation between marine outcrop area and marine diversity because they were using better quality data? The British data are well vetted and thorough; the western European outcrop area estimates are poor - the map square counts of Smith & McGowan (lines 511-512). These are quite a bit different from actual areas measured using GIS!

357-364: You find no correlation between global sea level and counts of tetrapod-bearing formations, and thereby reject the common cause model for either sampling or diversity of marine and non-marine tetrapods. Remarkable! An alternative view has been given by Butler and Benson, who compare parts of the marine tetrapod record and find a strong common cause model for the richest, onshore portions. In rejecting the common cause model as a control on diversity (line 361), you go beyond the earlier part of the paragraph, and contradict the main conclusion of your paper, that sea level drives diversity.

367: principle = principal

367-386: A series of time series model fits, showing sea level is generally the best predictor of diversity trends for both marine and terrestrial clades - AICc from SQS data (Table 1). However, you need to do these analyses in comparison with the raw diversity data also, surely?

Table 1:

For some reason, I could not find metrics in Table 1 for all tetrapods, or all-marine and all-non-marine - why are we not given these?

Supplement:

I pick out this justification of the use of collection counts and formation counts as sampling proxies from the Supplement (unnumbered page, unnumbered line): 'Irrespective of this variation, the fact that numerous studies have found significant correlations with additional, nonredundant proxies for the rock record^{40,55} (though see 38,39,56,57), suggests that fossil-bearing formation counts should represent an adequate proxy for the amount of rock record available for sampling.' Weasel words surely? Two references find correlation, four do not - weak evidence that any of these measures says anything about sampling. Further, as noted, you find little relation between any of these and the SQS-modified data in your analysis - QED.

In the considerable absence of figures in the main paper, I really think SI Fig. 1 should be in the main text. You have no evidence that this is not a more meaningful representation of what went on than the SQS-processed plots.

Reviewer #2 (Remarks to the Author)

This study presents an interesting analysis of biodiversity changes in tetrapods through the Jurassic/Cretaceous boundary using an impressive fossil data set. The authors investigate the magnitude of extinction across different marine and terrestrial clades while accounting for sampling biases and seek potential causes behind the inferred diversity patterns. Most clades appear to undergo either a substantial diversity loss through the J/K boundary or a marked turnover and reorganization of diversity. Through correlation analyses, sea level changes are identified as potentially playing a major role in determining the inferred diversity patterns. While I found the results of this study very interesting, I have some concerns about some of the methods implemented here and the presentation of the paper. Overall, I think the manuscript would benefit from a detailed assessment of the robustness of the results and an additional effort to improve the clarity of the text, especially to a broad audience of scientists outside the paleobiology field.

The whole section assessing diversity changes across the boundary is presented without discussing the statistical significance of the findings and the amount of uncertainty around the diversity estimates. Are the observed diversity changes significant? For instance, even under a constant birth death process at equilibrium (i.e. with origination rate equal to extinction rate) there can be quite substantial diversity variations, which are only linked to the stochasticity of the process. Do the observed diversity patterns deviate from these expectations? Estimating the true diversity of clades in deep time is clearly a challenge and I expect uncertainties to be quite large. Can such uncertainties be quantified, for instance with 95% confidence intervals around the curves in figure 1? Confidence intervals for diversity estimates are also important since the robustness of the SQS method applied here has been recently challenged through simulations (Hannisdal et al. 2016, <http://dx.doi.org/10.1101/043729>).

The effects of extrinsic factors are tested in terms of correlations with diversity patterns. However, if such factors had a mechanistic impact on tetrapods, we could expect tetrapods' diversification process (origination and extinction rates) to be directly correlated with environmental changes, rather than diversity patterns. Additionally, I wonder if competitive interactions and effects of trait evolution (p. 16 l. 390) couldn't be tested here as in Liow et al 2015 (doi: 10.1111/ele.12485), Slater 2015 (doi:10.1073/pnas.1403666111), Silvestro et al 2015 (doi: 10.1073/pnas.1502803112).

There are several technical terms in the manuscript that are likely going to be unclear to a non-specialist audience, such as 'Collection-based and formation based residual diversity estimates' (p. 5, second paragraph) and Lagerstätten effect (p. 10, l. 239). Also, what is a background rate in a scenario in which rates are changing through time (p. 8 l. 182)?

In Figures 1 and 2, it is not clear why the lines do not connect all the dots (i.e. when there a stage in between with no data).

The recent study by Starrfelt and Liow 2016 (doi: 10.1098/rstb.2015.0219) seems very relevant to the discussion of the findings in this study.

p. 2 l. 30: it is not clear what 'taxonomic and geographic constraints' are.

p. 8 l. 193: Did the authors estimate turnover rates?

p. 15 l. 369: The values reported as AICc are actually 'AICc weights'

Reviewer #3 (Remarks to the Author)

Dear authors,

Tennant & al. propose a thorough and statistically sound analysis of the diversity of tetrapods across the Jurassic-Cretaceous boundary (and beyond), highlighting the complexity of biodiversity patterns and the biases of the raw data. The evolution of diversity across this boundary has been the subject of some recent papers by Tennant & colleagues 1,2, but this paper provides a larger, deeper look at the general fluctuations of diversity, for a large series of groups and incorporating the worldwide record.

I strongly salute the cleverness of the authors, notably for their treatment of geological and anthropic biases and the soundness of the statistic treatments. The text is clear, a bit too long (some sections are repetitive between Results and Discussion chapters and could be condensed) and the references are appropriate. I have marked some typographic and grammatical errors and a series of comments and suggestions on a .pdf copy of the main text and a .docx file with supplementary methods, attached with the review. I reproduce and expand some of the most important comments below. I suggest publication after a revision.

There are three rather important points I would like to point out here and discuss with the authors; I am confident you will be able to work these out.

1. Lumping of bioevents and the possibly inflation of the importance of the Jurassic-Cretaceous boundary event(s)

In multiple occasions in the text, numerous Early Cretaceous events are incorporated into a set of global consequences arising from the Jurassic-Cretaceous boundary, such as the radiation of marine turtles or the Otong Java Plateau volcanism. Additionally, many marine groups have a poor earliest Cretaceous record and the authors thus compare Tithonian diversity with Valanginian or even Hauterivian diversity.

This appears to me as possibly far fetched, contributing to an impression of more severe extinction/turnover at the Jurassic-Cretaceous boundary than might have been. Indeed, many 15-20 million years windows in the fossil record are likely to contain a large series of temporally isolated events; lumping them gives a false impression of severity. Of course, it is possible that these changes in between the Tithonian and the Hauterivian are all located precisely at the J/K boundary, but the fossil record of a series of tetrapod groups is too imprecise to decipher between these two possibilities.

2. Reliance on shareholder subsampling as a universal method to correct palaeontological biases and the ichthyosaur problem

The development of SQS has been a huge improvement compared to other subsampling techniques, and has been used in a large series of recent analyses of past diversity. I think some discussion of the limits of the method should be provided, however.

It should be stated that you do not consider phylogeny to reconstruct the history of the groups analysed. One effect of this, is that your datums of origination will be the youngest boundary of all probable origination dates. As a direct consequence, you will necessarily have fewer groups crossing the boundary than actually happen, "worsening" the effect of the J/K in terms of extinction potential. It might also be interesting to state why you did not consider techniques to reconstruct ancient biodiversity based on phylogenies (time is an obvious one, of course!), and perhaps more importantly, why results obtained by SQS are better than those obtained by other methods - if they are. I am

referring here, notably, to ichthyosaurs as this is my group of expertise. I found it interesting to confront your data with mine, as I examined most Cretaceous specimens from Europe.

Notably, I am a bit worried about the ichthyosaur results. Hauterivian ichthyosaurs are extremely rare, only a handful of specimens that are diagnostic at the genus level or below are currently known, worldwide and a vast majority is restricted to a small area north of Hannover, Germany and two specimens in Speeton, UK. *Malawania anachronus* lineage cross the J/K boundary 3, but as its range is imprecise and spans the Hauterivian-Barremian boundary, I believe it was removed from the dataset; *Simbirskiasaurus* is actually Barremian in age 4. I am thus surprised that this poor record still passes the SQS quorum and is then used as a solid data point then directly compared to the subsampled Tithonian (which contains a series of lagerst ate, from many places of the world, unlike any Cretaceous stage) to evaluate the impact of the Jurassic-Cretaceous boundary on ichthyosaurs.

Are you sure these values can be trusted as an unbiased evaluation of ichthyosaur diversity across the Jurassic-Cretaceous boundary? I made other related comments in I170 p7 and I185-194 p8. Of course, these ichthyosaur issues are possibly an outlier and other groups considered have a better, more even record and treatment across the Jurassic-Cretaceous boundary.

You might consider putting less emphasis on groups with an inappropriate earliest Cretaceous record to evacuate these two first issues.

3. Invertebrate diversity as a proxy for marine food to find the drivers of the tetrapod diversity
I think there might be several issues with this rather simplistic assumption, notably because non-tetrapod vertebrates constitute a crucial food resource for marine tetrapods and diverse but temporally constrained and likely inedible groups such as Late Cretaceous hippuritoids probably bias the signal and thus, the correlation tests.

Good luck and best wishes,

Valentin Fischer

Rebuttal

Reviewer #1 (Remarks to the Author):

This is an enormously thorough piece of work, based on a great deal of effort in reviewing the literature and updating and correcting tetrapod data in the Paleobiology Database. The authors use generally accepted methods to process their data and to perform linear model fitting, and these approaches lead to their finding that most segments of their data can be explained best by sea level change – although some of the best passes are for clades where such an explanation is hard to comprehend, namely mammaliaforms, ornithischians, pterosaurs, sauropodomorphs, and theropods (Table 1).

Response: We thank the referee for their kind comments regarding our manuscript. The relationship between sea level and tetrapod diversity has been the subject of scientific scrutiny for many decades now. Indeed, some of the most recent analytical work on Mesozoic tetrapod diversity, such as Benson et al. (2011), Mannion et al. (2011), Butler et al. (2011) and Sakamoto et al. (2016), has examined the statistical relationship between tetrapod diversity and sea level, particularly in dinosaurs. The most easily recognizable biotic explanation for sea level influencing diversity in terrestrial tetrapod groups is that rising sea level leads to continental or landmass fragmentation, which alters the spatial distribution of habitats and affects the species-area relationship. Such fragmentation can be a potential driver for biological and reproductive isolation and speciation. As such, sea level can impact upon both speciation and extinction, and therefore is best compared with diversity, which is the standing product of these two rates. Therefore, our recovery of sea level as the primary agent influencing diversity throughout this time is consistent with previous studies, theoretically well-understood, and based on strong data and analytics. We have added the following to the discussion to clarify these points:

“The relationship between sea level and terrestrial diversity can best be explained via rising sea levels leading to landmass fragmentation. This alters the spatial distribution of near-shore habitats and affects the species-area relationship, which can lead to elevated extinctions. Such fragmentation can also be a potential driver for biological and reproductive isolation and allopatric speciation, the combination of which we would expect to see manifest in the diversity signal. However, evidence for these potential relationships between sea level, terrestrial diversity and sampling has remained elusive (Butler et al., 2011). As we find evidence for a positive correlation between sea level and diversity in multiple terrestrial clades, this suggests that allopatric speciation outweighs the species-area effect.”

However, I question the core claims of the paper for two reasons. First, because the results seem unexpected or hard to understand from fundamentals of ecology. For example, it is most unusual that diversity of marine crocodyliforms is controlled mostly by paleotemperature, whereas that of non-marine crocodyliforms is dependent on sea level. [I don't doubt the former, but very much query the

latter, which goes against decades of detailed work extending back to Markwick, whose work, and all subsequent work is uncited.

Response: A relationship between palaeotemperature and marine crocodyliforms was reported in a previous publication in Nature Communications (Martin et al., 2014). In two subsequent studies, we have almost entirely failed to replicate these results (Mannion et al., 2015; Tennant et al., 2016a), and instead found that sea level appears to be the primary external parameter influencing marine crocodyliform diversity. Indeed, we find similar results in the present study: “In the case of marine crocodyliforms, their subsampled diversity was driven by a combination of factors, including sea level, nutrient cycling and eustacy-influenced redox shifts”. We do not find the controlling relationship with palaeotemperature that is claimed by the reviewer. In Tennant et al. (2016a), we provided extensive discussion about how the diversity of ‘non-marine’ crocodyliforms appears to have been mediated by changes in sea level. Most importantly, most non-marine crocodyliforms around the Jurassic/Cretaceous transition were semi-aquatic or coastal forms, and therefore we would expect sea level to influence their habitats and therefore also their diversity (for the reasons mentioned above).

The only reason we do not cite the excellent and pioneering work by Markwick is due to the limits on the number of references allowed. We also note that much of Markwick’s work on crocodiles and palaeotemperature was primarily for the Late Cretaceous and Cenozoic, time periods which we do not cover in this manuscript.

Second, the methods veer between reasonable and unreasonable. The authors are aware of recent literature, and acknowledge many of the criticisms of popular methods in controlling sampling, but then, after partly accepting concepts such as common cause, redundancy, and the failure of the residuals method, proceed to defend and continue to use such highly criticised methods. The core method used here is SQS, but this is justified only by repeating assertions by Alroy as a justification, not for fundamental reasons, and not addressing published concerns that it does not do what is assumed. Throughout the MS, it is noted how all these different approaches sometimes correlate, and often do not, and yet they are all accepted as somehow pointing to sampling inadequacies [If cumulative metrics, residuals, and SQS were effective metrics of sampling, they should more often correlate and render the same results than not.] In relegating the raw, empirical data to the Supplement (SI Fig. 1) and basing the entire analysis on SQS-modified results, the authors repeat the approach of recent published research, but it could all be a house of cards. As they note, their SQS-modified results differ substantially from previous work, often by members of the same research group, who used this, and other methods of data-manipulation ('sampling correction'). Maybe SQS is more sensitive than they admit to modest changes in the input data (which it should not be, as it is meant to be correcting for gaps variable sampling).

Response: As far as we are aware, there has only been limited and brief criticism regarding the method (i.e., SQS) that we use, and it has been restricted to asides or the SI in a small number of publications. We would be grateful if the reviewer could highlight any such papers that we have missed. Much of the recent discussion in the relevant literature has focused on which proxies are appropriate to measure sampling in the fossil record, and we have discussed and accommodated these criticisms in depth

(particularly in the SI). For example, to account for the issue of potential redundancy, we use a 'higher level' proxy for sampling (i.e., tetrapod-bearing collections or formations), as recommended by Benton et al. (2013), but we do not use this as the basis to estimate 'residual diversity' due to recent criticisms of this method (e.g. Benton et al. 2013; Brocklehurst, 2015). We choose to analyse both empirical and subsampled diversity, the latter based on the SQS method, as well as two different and independent measures of extinction and origination rates. It is quite unfair to state that we are simply repeating Alroy as justification of using these methods, although we do credit him and refer to his work extensively as the author who designed these methods. Additionally we provide extensive detail of our methods in the SI that draws on much of Alroy's work, as typically when SQS has been utilized in the literature there has only been limited discussion of the method, and we felt that providing additional information would enhance the transparency and justification for our analyses. The fundamental reason for its use is that the fossil record contains structural bias, which we provide thorough discussion of (and evidence for) in the SI. We provide extensive justification for using this method both theoretically and empirically in the SI, as well as evidence of a range of additional studies that have employed it recently, and often with less discussion than we provide here. As reviewer #2 also notes, this method has recently been challenged by Hannisdal et al. (in review), but this work currently only exists as a non-peer-reviewed preprint, and it is perhaps unfair to expect us to take into account unpublished work at this stage.

Nonetheless, the major criticism of SQS within that preprint is that it is sensitive to changes in the species distribution curve, and subsampled diversity estimates using this method appear to correlate with the evenness of abundance distributions as defined by Pielou's J . However, we do not view that as a weakness, but as a strength – changing the species sample pool should be reflected in changes in subsampled diversity, especially if this sampling perturbation effects the estimated evenness. And this is precisely what happens with SQS based on the evenness estimates using Good's u . Whether or not dismissing an otherwise widely used and statistically rigorous method can be based on the assumptions around one measure of evenness, remains to be analytically tested. Irrespective of these potential issues, which are beyond the scope of this paper, SQS diversity has been shown repeatedly to recover a reliable diversity signal (Alroy, 2010; Mannion et al., 2015; Nicholson et al., 2015), and therefore its use here, and elsewhere, is warranted until it can be demonstrated to be a weak estimator of true diversity and based on skewed or incorrect calculations of evenness. The fact that the method has been employed across a broad spectrum of palaeontological analyses (e.g., Hannisdal and Peters, 2011; Nicholson et al., 2015) suggests that it is generally an accepted valuable analytical method, and we await published criticism of the method before dismissing it. We do not believe that several comments in a non-peer reviewed manuscript should provide the basis for undermining a widely accepted and used methodology.

While we demonstrate consistently throughout the manuscript that raw, empirical diversity is a poor measure of true diversity due to it containing a strong sampling signal, as numerous studies have found, we have moved the figure illustrating raw diversity from the supplementary material to the main manuscript to highlight the differences, as mentioned, and we agree that it is good to show the dataset in this form.

The reviewer is correct that some of our results vary compared to previously published results, but not those using SQS. The only time we see discrepancies is when we vary the time bin duration (between stages and 10 million year time bins), which we would expect as the binning method clearly controls the shape of the species sample pool. Our results vary somewhat to those that were based on residual diversity estimates, a method which has been shown to perform relatively poorly at different scales (Brocklehurst, 2015), and especially when sampling levels are low, such as during the earliest Cretaceous. We do not employ the residuals method here. However, other recent studies have shown that SQS diversity correlates with residual diversity (Smith et al., 2012) and phylogenetic diversity estimates (Tennant et al., 2016a), and therefore it is most likely capturing a similar sampling signal.

Some specific points

158-160: 'Previous studies of tetrapods have failed to find a correlation between time bin length and raw diversity, suggesting that stage level bins are appropriate for diversity studies.' Does it? The absence of a correlation simply means there is no correlation, and does not remove the possibility that short bins sample less than long bins. With such patchy sampling as in the J-K interval, you cannot tell that all sorts of confounding factors might not be occurring. Dividing diversity by bin length is still not excluded. Secondly, this statement ignores a point made earlier that a bin of any length might confound numerous non-contemporaneous faunas and lose rapid fluctuations in diversity. The similarity of bin length (line 162) is the more useful point.

Response: The possibility that shorter time bins might sample less diversity than longer bins is one of the reasons why subsampling protocols like SQS and even classical rarefaction are so powerful, in that they alleviate biases associated with different sample sizes. It is almost ubiquitous that sample sizes will differ based on some factor – the question is how we deal with them, and our answer is through application of a fair subsampling protocol. Regarding the potential confounding factor of grouping non-contemporary collections within bins, this is a problem that we accept as a limitation for all diversity studies based simply on the resolution of the geological and fossil records. However, one strength of our study is that it examines the sensitivity of our conclusions to different time-binning strategies. While time-bin duration does make a difference, many of our conclusions are supported by the analyses based on both stage-level and 10 million year time bins, and we are explicit on those occasions where discrepancies occur. Furthermore, simply dividing diversity by bin length is an arbitrary solution to this problem, and makes the assumption that diversity will linearly increase with bin length. We do not make this assumption, as it has the potential to introduce error into our analyses.

274-277: Do you have any evidence that comparisons of raw diversity and raw collection counts or formation counts say anything about sampling? This information is largely relegated to the Supplementary Materials, but you should not refer to collection counts or formation counts as sampling metrics, as they likely are not. They are simply counts of collections and formations that mirror the counts of specimens and species as a result of a common history of data accumulation (redundancy).

Response: This information is within the SI due to constraints on manuscript size, and we wished to present as many new results and associated discussion as possible. The discussion about the interplay between collection and formation counts and raw diversity has exploded recently with analyses at a range of different taxonomic, temporal, and geographic scales. We discuss many of these and reach conclusions about how best to treat these 'sampling metrics'. If we had not included these data, our manuscript would have been dismissed outright for not considering sampling metrics, and therefore we concluded that it is best to include analysis of them at different levels. We discuss the implications of these data in the context of redundancy, finding that in almost all cases raw empirical diversity can be explained by the redundancy hypothesis, but that this association is broken down once we apply subsampling. Therefore we agree with the referee that raw taxonomic diversity is often redundant with 'sampling metrics' (especially those not using a higher taxonomic level, such as tetrapod-bearing collections, as the proxy), and this is indeed why we made the decision to exclude these data from the main manuscript, contrary to the recommendations above. Note that we have now included the 'raw' diversity data in the main MS.

285-301: Note that actual rock exposure (= availability of rock for collection) may be entirely unrelated to outcrop estimates, especially in heavily agricultural temperate areas like Europe.

Response: We note in the SI that exposure area and outcrop area are likely not correlated based on the detailed work of Dunhill and colleagues: "The use of outcrop area as a sampling proxy remains questionable due to inconsistency in its relationship with the amount of rock available from which to sample fossils (i.e., exposure area)." However, measuring global or continent-level rock exposure area would require a global collaboration and the compilation of geological data that is not currently available at the temporal level required, and is therefore beyond the scope of this study. We are also very explicit in our treatment of outcrop area as a proxy at a regional level, in addition to formation counts, as proxies for the availability of the rock record for sampling, and recognise its limitations. There are also limitations to using current exposure estimates for diversity studies. For example, in heavily industrialised areas such as Europe, outcrop area represents the total pool of opportunities to sample fossils over historical time, a factor that is not captured in current exposure estimates. Therefore exposure estimates fail to account for historical variation in sampling availability through changes in exposure at, for example, quarrying sites, eroding cliffs, and road cuttings. As such, neither outcrop nor exposure will perfectly capture all aspects of sampling opportunity.

302-304: 'On a global level, both geological and anthropogenic sampling appear to control raw taxonomic diversity, but this is alleviated when subsampling is applied, as observed from the switch from almost universally significant positive correlations to no correlations.' You have provided no evidence for this. The fact that different collections, formations, and rock area metrics correlate variously with diversity perhaps suggests all those 'sampling metrics' are poor estimators of sampling failure. The best estimator is surely to identify the minimally known unknowns, in other words, Lazarus taxa and ghost ranges from phylogenies (see Smith 1994, 2007 etc). These point to bins with known missing lineage records - it's not great, but it's a start. All the other stuff is guesswork and assertion since all those signals are either mutually entwined (= redundant) or themselves probably more patchy and incomplete (e.g. rock area/ volume) than the raw diversity data.

Response: We provide substantial evidence in the supplementary data files for our correlation and model fitting results. As mentioned above, we note that the relationship between each of these ‘sampling metrics’ and raw diversity is likely the product of redundancy. We tested this with raw diversity at numerous different taxonomic levels using potentially redundant and non-redundant proxies, and yet find it to be a consistent pattern. If this pattern was not so ubiquitous, then we would be inclined to agree with the referee that our proxies most likely represent poor sampling estimates. In order to reduce the effect of redundancy, we applied a higher level set of proxies (i.e., at the tetrapod-level) as has been recommended by previous workers (e.g. Benton et al., 2011, 2013), but still find consistently strong correlations between these and raw diversity. We agree that there is certainly a benefit in estimating phylogenetic diversity and comparing this with our results. However, creating a global tetrapod phylogeny at the resolution of the dataset used here would take a monumental global effort – no one has yet produced such a large phylogeny. Furthermore, phylogenies exclude many (and often most) taxa for reasons pertaining to specimen completeness and access, as well as time, and thus phylogenetic diversity estimates (PDEs) are based on only a small sample of known (observed) diversity. PDEs also have other issues, detailed in Lane et al. (2005). In any case, there is evidence that the approaches taken in our study (i.e. SQS) and PDEs often produce similar conclusions about past diversity. For example, our previous comparisons between phylogenetic diversity estimation and subsampled crocodyliform diversity using SQS suggest a close correlation between the two (Tennant et al., 2016a).

304-307: Interesting - so SQS has additional properties, not only dealing with patchiness of collecting and recording into the PBDB, but also geological bias? This is weak.

Response: This is an unexpected result of our analyses. Indeed, we state explicitly: “We urge caution in the interpretation of non-significant results as evidence for no relationship, this shift in correlation strength occurs in almost every taxonomic group, independently of their sampling histories and overall diversity patterns. These results collectively suggest that SQS is an adequate method to account for fossil record bias, as opposed to scrambling a common underlying signal influencing both sampling and diversity.” Therefore as opposed to being “weak”, the pattern is almost ubiquitous, and we suggest a possible reason for this: “This utility of SQS most likely occurs because the relationship between regional collection counts and outcrop area is consistently strongly positively correlated in the non-marine and marine realms.”, the results of which are provided in SI 6. As far as we are aware, this is the first time that such an association has been made, and we welcome additional research into this unexpected property of subsampling.

314-315: 'The exception to this is the marine record of Europe, in which outcrop area appears to be independent of raw and subsampled diversity.' Interesting. So the record that geologists would likely a priori predict to be best of all - heavily sampled Europe, and European marine rocks? - turns out worst. Maybe the battery of tests is not doing what you think it is.

Response: The record is not itself ‘worst’. It is the relationship between diversity and a proxy for the rock record that is worst. Indeed, because the sampling history of Europe is much ‘better’ relative to the rest of the world, we might expect such a lack of correlation to exist.

315-317: 'Furthermore, outcrop area is independent of any other regional sampling metric, which probably reflects Europe having a more intensive sampling history compared to the rest of the world.' Surprising result - surely this should make you think twice. It is more likely that the regional 'sampling' metrics are not sampling metrics at all, but redundant correlates of diversity, or meaningless variable numbers. If your assumptions and methods don't work in the best sampled portion of the record, I'd be worried about all the prior assumptions.

Response: If there was redundancy between European outcrop area and sampling, then we would expect a correlation, which we don't find. We haven't made any assumptions about whether these proxies are redundant or not – we have performed these analyses to test precisely that. At this scale, based on our data, we find that European outcrop area and our other regional sampling metrics are not correlated. It is not necessarily that they are then meaningless numbers, but the more likely explanation is simply that they are both measuring different aspects of the geological record and sampling. As the reviewer noted above, just because we do not find a correlation between time bin length and diversity, this does not mean there is no relationship between the two, which logically contradicts the statement here that the non-correlation means there is no meaningful relationship.

321-324: 'suggesting that increasing outcrop area (i.e., geological sampling) has little to no effect on the overall evenness and structure of tetrapod sampling for reconstructing diversity, irrespective of whether this means there are more opportunities to sample (collections) or not.' Does this then suggest that you might be best to work with the empirical signal rather than a data manipulation of uncertain effectiveness?

Response: No, as the empirical data are structured in a manner that is strongly controlled by our sampling metrics. Our analytical protocol is not of uncertain effectiveness, as it has been demonstrated to be a good estimator for true diversity repeatedly and at different scales. All our results mean is that the shape of the species abundance curve in North America is non-dependent on outcrop area. Empirical diversity does not even factor in this distribution, and as has been repeatedly demonstrated in this manuscript and numerous others analytically for the last 40 years: empirical diversity is a poor, biased estimate of true diversity. The key debate among nearly all workers in the field of estimating past diversity is not whether the fossil record can be read literally, as it has been repeatedly demonstrated that this is a poor practice; rather, it is which of the various correction approaches are most effective. Indeed, the reviewer stated previously that phylogenetic methods are required to estimate diversity more accurately, but contradicts this by suggesting use of empirical diversity data here.

327-330: 'However, subsampled diversity, as a more accurate estimate of true diversity, is independent of higher taxonomic-level sampling metrics (i.e., MBFs/TBFs and MBCs/TBCs) at a global level (SI 6), meaning that we can reject the hypothesis that 'true' diversity controls sampling, and so does not control our sampling metrics.' Is there any evidence that SQS does all this? We only have Alroy's assertions of its properties, and nobody has provided fundamental evidence why. As you noted earlier, SQS was designed to deal only with human sampling bias, not geological bias, and surely geological patchiness is rampant throughout your J-K records (e.g. massive facies switch in Europe from marine to nonmarine at the J-K boundary), and it reduces the data to a much flattened-

out version of the raw signal. SQS does not provide an absolute signal, merely a relative signal, although that's fine for detecting rises and falls, but not really magnitudes of those rises and falls. SQS is likely sensitive to the exact bins chosen and the exact figures estimated - for example, Alroy's SQS estimates of global diversity are sensitive to the current state of the PBDB - add another tranche of data, and it alters massively. It shouldn't do this if it is truly accounting for sampling.

Response: The evidence that SQS does this is presented in the current manuscript, in the sentence the referee quotes. The data supporting this is all in SI 6. This is not something which Alroy originally even mentioned in his series of manuscripts describing and employing SQS, as he largely disregards any sort of 'geological sampling bias' in favour of approaching the problem through subsampling. The referee is also correct in pointing out that there are major geological shifts across the J/K boundary (reviewed to some extent in Tennant et al., 2016b). However, as we note in our manuscript, SQS appears to deal with the variation this imposes on sampling, based on our geological proxies. This occurs for both marine and non-marine tetrapod groups. Our raw data is actually much flatter (see Supplementary Figure 1, now Figure 1) than the shape of our subsampled data.

SQS captures relative magnitude, which is arguably a more valuable measurement seeing as that is what we are asking of our data – the relative changes between time bins. Absolute measures of diversity are difficult based on the fossil record, and recent attempts at estimating this are in their infancy (e.g., Starrfelt and Liow, 2016). Estimating absolute diversity is not one of the aims of the present study, only relative change(s) across the boundary, and therefore SQS is sufficient.

The statement that the shape of the richness curve should not alter when new data is added is incorrect, as this is exactly what we would want of a method that accounts for variation in the species sampling pool. We also provide analyses at both the stage level and 10 million year time bin level and, as one might expect, we recover quite different results, as these binning schemes drastically alter the sample pool. One would expect, and indeed want, resulting richness to vary as new data is added. If a method produced exactly the same results if we added 'another tranche of data', then we should be concerned that it is not capturing any sort of sampling signal or changes in the taxon abundance distribution. This is by no means a fault of SQS, but due to performing analyses on incomplete data. Our dataset is one of the most comprehensive for tetrapods ever built as part of a global collaboration over many years. It has been thoroughly checked over for accuracy and to make sure it is an accurate and comprehensive representation of the current published literature. This is less a critique of SQS, and more a note of caution about using incomplete/immature datasets, which we completely agree with.

330-334: 'Nor are our regional rock record metrics and subsampled diversity estimates the product of 'redundancy', a similar conclusion to that reached by Upchurch et al. for North America and Europe, which further implies that our potentially redundant proxies (i.e., formation and collection counts) are capturing a genuine regional sampling signal.' That's a bold assertion. Just because you say it (and other people have said it) does not mean it is true - you have to provide evidence. If SQS is the wunder-metric, and SQS does not relate to any of the cumulative collections/ formations/ rock area metrics, then your statement is more likely false. [The following sentence is then redundant.]

Response: The evidence is provided in SI 7 to support these conclusions. We provide maximum likelihood fitting results and standard pairwise correlation tests to demonstrate that in North America, our proxy for the rock record is independent of the few groups for which we were able to recover a roughly continuous subsampled diversity estimate. We also note that SQS is not a metric in itself, but a method for estimating diversity while accounting for uneven sampling.

351-356:... or maybe Dunhill et al. found a strong correlation between marine outcrop area and marine diversity because they were using better quality data? The British data are well vetted and thorough; the western European outcrop area estimates are poor - the map square counts of Smith & McGowan (lines 511-512). These are quite a bit different from actual areas measured using GIS!

Response: Undoubtedly the data collected and published by Dunhill and colleagues in a series of papers is of higher quality due to the scale of the data and mode of collection (besides the concerns we have noted above). However, here our work focuses on continent-level analyses, and no such GIS compilations at this spatial (and suitable temporal) scale have yet been constructed. We have used the best data available to us at the time, and look forward to when our field has advanced sufficiently that we can conduct analyses similar to the approach of Dunhill and colleagues on a continental/global scale. We also choose not to make judgements about the work of Smith and McGowan just because it uses an alternative model to Dunhill and colleagues.

357-364: You find no correlation between global sea level and counts of tetrapod-bearing formations, and thereby reject the common cause model for either sampling or diversity of marine and non-marine tetrapods. Remarkable! An alternative view has been given by Butler and Benson, who compare parts of the marine tetrapod record and find a strong common cause model for the richest, onshore portions. In rejecting the common cause model as a control on diversity (line 361), you go beyond the earlier part of the paragraph, and contradict the main conclusion of your paper, that sea level drives diversity.

Response: The work by Butler and Benson was largely based on using residual diversity estimates. As mentioned previously by the reviewer and elsewhere, these might not be as accurate as those obtained using a subsampling protocol, which might explain the differences with the present study. We also did not differentiate between near-shore and offshore taxa or environments, as these authors did. The main conclusion of our study is that sea level regulates diversity. This is distinct from the common cause hypothesis which states that sea level drives both diversity and sampling. Thus we can logically support the first conclusion while also arguing against the common cause without the purported contradiction.

367: principle = principal

Response: Corrected.

367-386: A series of time series model fits, showing sea level is generally the best predictor of diversity trends for both marine and terrestrial clades - AICc from SQS data (Table 1). However, you need to do these analyses in comparison with the raw diversity data also, surely?

Response: We have done this, and these results are fully provided in SI 7 alongside the comparisons with subsampled diversity. These additional analyses do not affect the main conclusions of our paper, as we find that it is subsampled diversity that is most strongly associated with sea level.

Table 1:

For some reason, I could not find metrics in Table 1 for all tetrapods, or all-marine and all-non-marine - why are we not given these?

Response: These are all supplied in SI 7. This is noted in the table caption.

Supplement:

I pick out this justification of the use of collection counts and formation counts as sampling proxies from the Supplement (unnumbered page, unnumbered line): 'Irrespective of this variation, the fact that numerous studies have found significant correlations with additional, nonredundant proxies for the rock record^{40,55} (though see ^{38,39,56,57}), suggests that fossil-bearing formation counts should represent an adequate proxy for the amount of rock record available for sampling.' Weasel words surely? Two references find correlation, four do not - weak evidence that any of these measures says anything about sampling. Further, as noted, you find little relation between any of these and the SQS-modified data in your analysis - QED.

Response: We do not elect to make decisions about the validity of proxies based on the largely arbitrary number of references provided in the text. These examples of studies were provided simply to demonstrate that there is no current consensus about the appropriateness of 'sampling proxies'. These papers were cited to avoid drawing exclusively on tetrapod-based studies, which have by and large favoured the use of these proxies. We have simply replaced 'should' with 'could' to reflect this.

In the considerable absence of figures in the main paper, I really think SI Fig. 1 should be in the main text. You have no evidence that this is not a more meaningful representation of what went on than the SQS-processed plots.

Response: We elected to not include many figures due to length constraints in the journal. While we have shown that raw taxonomic data is not a good measure of diversity, we have now included this in the main manuscript as Figure 1 in order to highlight the differences between this and the subsampled estimates provided in Figures 2 and 3.

Reviewer #2 (Remarks to the Author):

This study presents an interesting analysis of biodiversity changes in tetrapods through the Jurassic/Cretaceous boundary using an impressive fossil data set. The authors investigate the magnitude of extinction across different marine and terrestrial clades while accounting for sampling biases and seek potential causes behind the inferred diversity patterns. Most clades appear to undergo either a substantial diversity loss through the J/K boundary or a marked turnover and reorganization of diversity. Through correlation analyses, sea level changes are identified as potentially playing a major role in determining the inferred diversity patterns. While I found the results of this study very interesting, I have some concerns about some of the methods implemented here and the presentation of the paper. Overall, I think the manuscript would benefit from a detailed assessment of the robustness of the results and an additional effort to improve the clarity of the text, especially to a broad audience of scientists outside the paleobiology field.

The whole section assessing diversity changes across the boundary is presented without discussing the statistical significance of the findings and the amount of uncertainty around the diversity estimates. Are the observed diversity changes significant? For instance, even under a constant birth death process at equilibrium (i.e. with origination rate equal to extinction rate) there can be quite substantial diversity variations, which are only linked to the stochasticity of the process. Do the observed diversity patterns deviate from these expectations? Estimating the true diversity of clades in deep time is clearly a challenge and I expect uncertainties to be quite large. Can such uncertainties be quantified, for instance with 95% confidence intervals around the curves in figure 1? Confidence intervals for diversity estimates are also important since the robustness of the SQS method applied here has been recently challenged through simulations (Hannisdal et al. 2016, <http://dx.doi.org/10.1101/043729>).

Response: At the present, SQS does not produce confidence intervals. While implementing a bootstrapping procedure to obtain these is possible, and relatively simple in the R version of SQS, we have found in previous studies that this version produces different and most likely less reliable results than that obtained using SQS implanted in Perl, as conducted here. However, when we implanted this procedure for SQS in R (Tennant et al., 2016a), the standard deviations were extremely small (as one might expect with our generally small sample sizes). Therefore, although we agree that this would be a valuable addition in the future, it should not detract from our overall results, and we avoid commenting on significance, as with all previous studies employing SQS, and reserve our comments to the means produced using the SQS algorithm. As we responded to the first reviewer, while the Hannisdal et al. (in review) manuscript seems to be important, we believe that the supposed ‘weaknesses’ of SQS briefly outlined in that manuscript (which currently exists only as a non-peer reviewed preprint) are not actually weaknesses but fundamental properties of how a ‘good’ method should respond to variation in the sample pool. We look forward to seeing the final version of that paper published and a more detailed investigation of the properties of SQS, but we are hesitant to discard our results based on currently unpublished research (with all due respect to the authors of that paper).

The effects of extrinsic factors are tested in terms of correlations with diversity patterns. However, if such factors had a mechanistic impact on tetrapods, we could expect tetrapods' diversification process (origination and extinction rates) to be directly correlated with environmental changes, rather than diversity patterns. Additionally, I wonder if competitive interactions and effects of trait evolution (p. 16 l. 390) couldn't be tested here as in Liow et al 2015 (doi: 10.1111/ele.12485), Slater 2015 (doi:10.1073/pnas.1403666111), Silvestro et al 2015 (doi: 10.1073/pnas.1502803112).

Response: We do not test extinction and speciation rates with our environmental data, as these are measured at different 'points' of each bin. Our environmental data are in-bin data, whereas the rates are instantaneous boundary-crossing data. Furthermore, diversity captures both elements of speciation and extinction in one, and therefore captures more information about the overall processes.

Future research on tetrapod macroevolutionary patterns through this interval should certainly explore the effects of competition and phenotypic change; however, these are large-scale studies in themselves requiring substantial additional data collection and a global tetrapod phylogeny, which does not yet currently exist, and we hope in future to use the present research as the basis for further investigation along these lines. We mention this in the text: "Where we recover relatively lower AICc weights, this indicates that additional parameters that we did not analyse here, such as post-extinction opportunism or competitive displacement, or passive aspects of trait evolution, might also have played a significant role in affecting global diversity patterns for certain groups." As such, we feel that this is beyond the scope of the present study.

There are several technical terms in the manuscript that are likely going to be unclear to a non-specialist audience, such as 'Collection-based and formation based residual diversity estimates' (p. 5, second paragraph) and Lagerstätten effect (p. 10, l. 239). Also, what is a background rate in a scenario in which rates are changing through time (p. 8 l. 182)?

Response: These residual methods are discussed in much more detail in the supplementary methods, but we have now indicated this in the text for further clarity. For the Lagerstätten effect, we have added: "(i.e., episodes of greatly enhanced fossil record preservation)". We have also clarified the sentence on background rates: "rate of other intervals during the Jurassic".

In Figures 1 and 2, it is not clear why the lines do not connect all the dots (i.e. when there a stage in between with no data).

Response: This is where subsampling does not return a result in a bin due to a relatively high quorum and poor sampling. We chose not to extrapolate between points and instead explicitly show which intervals remain relatively poorly sampled. We have added this to the figure legends for additional clarification: "Where gaps in the curve exist, this is due to poor sampling and failure to adequately recover a subsampling diversity estimate."

The recent study by Starrfelt and Liow 2016 (doi: 10.1098/rstb.2015.0219) seems very relevant to the discussion of the findings in this study.

Response: We agree that this is an important study for a multitude of reasons. It's exclusion from our MS was due to the fact that it was published after our original submission of this manuscript. While a full analysis of the new method proposed in that paper, and particularly with respect to SQS, is beyond the scope of the present study, we have now incorporated their results into our discussion. We have added the following to the main manuscript text:

“Our results contrast with those recently obtained using the novel TRiPS method (Starrfelt and Liow, 2016), which found that each of the three major dinosaur clades did not suffer a diversity loss over the J/K boundary when simultaneously calculating both sampling rate and richness. The relative performance of TRiPS to SQS is beyond the scope of this manuscript, but is a factor that requires future investigation.”

p. 2 l. 30: it is not clear what 'taxonomic and geographic constraints' are.

Response: We have changed this to the following: “but taxonomic selectivity and an apparent geographic constraint to Europe”. Additional information is provided in the references we cite here, including a comprehensive review of the Jurassic/Cretaceous transition by the present authors.

p. 8 l. 193: Did the authors estimate turnover rates?

Response: We did not, but we did estimate speciation and extinction rates independently.

p. 15 l. 369: The values reported as AICc are actually 'AICc weights'

Response: We thank the referee for pointing out our mistake here – we have updated the text throughout to clarify this.

Reviewer #3 (Remarks to the Author):

Dear authors, Tennant & al. propose a thorough and statistically sound analysis of the diversity of tetrapods across the Jurassic-Cretaceous boundary (and beyond), highlighting the complexity of biodiversity patterns and the biases of the raw data. The evolution of diversity across this boundary has been the subject of some recent papers by Tennant & colleagues 1,2, but this paper provides a larger, deeper look at the general fluctuations of diversity, for a large series of groups and incorporating the worldwide record. I strongly salute the cleverness of the authors, notably for their treatment of geological and anthropic biases and the soundness of the statistic treatments. The text is clear, a bit too long (some sections are repetitive between Results and Discussion chapters and could be condensed) and the references are appropriate. I have marked some typographic and grammatical errors and a series of comments and suggestions on a .pdf copy of the main text and a .docx file with

supplementary methods, attached with the review. I reproduce and expand some of the most important comments below. I suggest publication after a revision. There are three rather important points I would like to point out here and discuss with the authors; I am confident you will be able to work these out.

1. Lumping of bioevents and the possibly inflation of the importance of the Jurassic-Cretaceous boundary event(s). In multiple occasions in the text, numerous Early Cretaceous events are incorporated into a set of global consequences arising from the Jurassic-Cretaceous boundary, such as the radiation of marine turtles or the Otong Java Plateau volcanism. Additionally, many marine groups have a poor earliest Cretaceous record and the authors thus compare Tithonian diversity with Valanginian or even Hauterivian diversity. This appears to me as possibly far fetched, contributing to an impression of more severe extinction/turnover at the Jurassic-Cretaceous boundary than might have been. Indeed, many 15-20 million years windows in the fossil record are likely to contain a large series of temporally isolated events; lumping them gives a false impression of severity. Of course, it is possible that these changes in between the Tithonian and the Hauterivian are all located precisely at the J/K boundary, but the fossil record of a series of tetrapod groups is too imprecise to decipher between these two possibilities.

Response: When these events are referenced, it is with respect to events around the J/K transition, as opposed to the boundary itself. We have been very careful in this MS, and in our previous publications on this topic, to note events explicitly at the J/K boundary, and those that occur either side of the boundary. We do this in order to note that it is the J/K transition that is of broader interest, and not necessarily just those events directly at the boundary. We do compare diversity over several stages (and using two time bin schemes) in order to understand more broadly what is happening to different groups throughout the transition, and note (especially with marine groups) were poor sampling means that we cannot directly interpret patterns over the J/K boundary itself. When we discuss boundary-events, it is specifically with respect to those at the Tithonian-Berriasian transition, and we are careful to note this throughout. We have checked through the text to make sure that our wording is consistent on this matter.

2. Reliance on shareholder subsampling as a universal method to correct palaeontological biases and the ichthyosaur problem. The development of SQS has been a huge improvement compared to other subsampling techniques, and has been used in a large series of recent analyses of past diversity. I think some discussion of the limits of the method should be provided, however.

Response: It should be stated that you do not consider phylogeny to reconstruct the history of the groups analysed. One effect of this, is that your datums of origination will be the youngest boundary of all probable origination dates. As a direct consequence, you will necessarily have fewer groups crossing the boundary than actually happen, "worsening" the effect of the J/K in terms of extinction potential. It might also be interesting to state why you did not consider techniques to reconstruct ancient biodiversity based on phylogenies (time is an obvious one, of course!), and perhaps more importantly, why results obtained by SQS are better than those obtained by other methods - if they are. I am

referring here, notably, to ichthyosaurs as this is my group of expertise. I found it interesting to confront your data with mine, as I examined most Cretaceous specimens from Europe.

In a previous publication of ours on crocodyliforms, we did compare the results of a phylogenetic-based diversity analysis with that obtained using SQS, finding the results to be quite similar overall with respect to the J/K transition. As noted in our response to reviewer 1, doing this for all tetrapod groups would require a large-scale collaboration that is beyond the scope of the present study. Furthermore, these methods are unable to account for relative sampling differences as mentioned, which is why we elected to use a three-pronged approach of estimating diversity, origination, and extinction. Using phylogenetic methods actually can have the adverse effect of ‘dampening’ extinctions as it skews estimates by pulling back origination times across intervals of low sampling (Wagner, 2000). We have added a section on why we have primarily elected to use SQS to our supplementary methods, and included the following points of discussion:

“Numerous methods have been developed in order to account for this heterogeneity, including model-based and phylogenetic methods. The robustness of modelling approaches, in particular estimates of ‘residual’ diversity, has recently been questioned, and might not reproduce faithful estimates of diversity (Brocklehurst, 2015). While phylogenetic methods might be superior for estimation of extinction and origination rates, conducting such analyses for all tetrapods would require a global collaboration effort that is beyond the scope of the present study. Additionally, phylogenetic approaches can have the adverse effect of imposing an asymmetry in analyses by only correcting origination times and not extinction times, and are also highly sensitive to changes in phylogenetic hypotheses and often the selective inclusion of only well-known taxa.”

“One recent study (available as a pre-print) has questioned the applicability of SQS due to its sensitivity to changes in the species-abundance distribution (Hannisdal et al., 2016). Other methods such as TRiPS have been recently developed (Starrfelt and Liow, 2016), but have not been shown to consistently outperform diversity estimation compared to SQS as of yet.”

Notably, I am a bit worried about the ichthyosaur results. Hauterivian ichthyosaurs are extremely rare, only a handful of specimens that are diagnostic at the genus level or below are currently known, worldwide and a vast majority is restricted to a small area north of Hannover, Germany and two specimens in Speeton, UK. Malawania anachronus lineage cross the J/K boundary 3, but as its range is imprecise and spans the Hauterivian-Barremian boundary, I believe it was removed from the dataset; Simbirskiasaurus is actually Barremian in age 4. I am thus surprised that this poor record still passes the SQS quorum and is then used as a solid data point then directly compared to the subsampled Tithonian (which contains a series of lagerstätte, from many places of the world, unlike any Cretaceous stage) to evaluate the impact of the Jurassic-Cretaceous boundary on ichthyosaurs. Are you sure these values can be trusted as an unbiased evaluation of ichthyosaur diversity across the Jurassic-Cretaceous boundary? I made other related comments in I170 p7 and I185-194 p8. Of course, these ichthyosaur issues are possibly an outlier and other groups considered have a better, more even record and treatment across the Jurassic-Cretaceous boundary. You might consider putting less

emphasis on groups with an inappropriate earliest Cretaceous record to evacuate these two first issues.

Response: We refer the Editor to our previous responses regarding the use and appropriateness of SQS, but herein provide additional information directly relevant to ichthyosaurs. One of the great things about SQS is that it tells us when sampling is good enough to pass a 'quorum'. For the earliest Cretaceous, sampling of ichthyosaurs is very poor, as noted, and we fail to return a subsampled diversity estimate. Those taxa and occurrences which are not resolved well enough temporally to have their entire ranges constrained within a single time bin are excluded from our analyses. In the Hauterivian-Barremian interval, while sampling is still quite poor, coverage is still sufficient based on Good's u ($u = 0.63$) to return a subsampled estimate with a quorum of 0.4. This is based on, as mentioned, a single German occurrence, two English occurrences (all three of the same genus), and a single Russian occurrence of a second genus (i.e., a singleton). The resulting diversity estimate is still extremely low (1.5 for the Hauterivian) based on this, which is less than half of that for the Tithonian. This is the only subsampled diversity estimate we recover for the whole Early Cretaceous interval for ichthyosaurs prior to the Aptian (1.25) based on this overall poor sampling regime, as the reviewer is correct to point out. Indeed, as mentioned by the reviewer, there are a series of marine Lagerstätten in the Late Jurassic of Europe. The whole purpose of applying a fair subsampling protocol like SQS is that it accounts for variations in sampling that result from potential bias from anomalously large collections.

3. Invertebrate diversity as a proxy for marine food to find the drivers of the tetrapod diversity

I think there might be several issues with this rather simplistic assumption, notably because non-tetrapod vertebrates constitute a crucial food resource for marine tetrapods and diverse but temporally constrained and likely inedible groups such as Late Cretaceous hippuritoids probably bias the signal and thus, the correlation tests.

Response: While we are careful to note that this is a very coarse proxy for food resource availability, we think that the reviewer is probably correct in noting that this is too simplistic. SQS diversity is probably a more accurate reflector of just that – the diversity of marine invertebrates through time, in this case. Our inclusion of this variable was part of an exploratory analysis, and we are careful not to over-interpret the single correlation that we recover with it. We have removed the following: "which we apply as a coarse proxy for potential food resources for marine groups." and " , or, perhaps unexpectedly, inversely to shifts in the availability of food resources", to reflect this.

Good luck and best wishes,

Valentin Fischer

1. Tennant, J. P., Mannion, P. D., Upchurch, P., Sutton, M. D. & Price, G. D. Biotic and environmental dynamics through the Late Jurassic - Early Cretaceous transition : evidence for protracted faunal and ecological turnover. *Biol. Rev.* doi:10.1111/brv.12255
2. Tennant, J. P., Mannion, P. D. & Upchurch, P. Environmental drivers of crocodyliform extinction across the Jurassic/Cretaceous transition. *Proc R Soc B* 283, 20152840- (2016).
3. Fischer, V. et al. A basal thunnosaurian from Iraq reveals disparate phylogenetic origins for Cretaceous ichthyosaurs. *Biol. Lett.* 9, 1-6 (2013).
4. Fischer, V. et al. Simbirskiasaurus and Pervushovisaurus reassessed: implications for the taxonomy and cranial osteology of Cretaceous platypterygiine ichthyosaurs. *Zool. J. Linn. Soc.* 171, 822-841 (2014).

Reference list

- Alroy J (201) The shifting balance of diversity among major marine animal groups. *Science.* 329: 1191-1194
- Benson R B, Butler R J, Lindgren J, Smith A S (2009) Mesozoic marine tetrapod diversity: mass extinctions and temporal heterogeneity in geological megabiases affecting vertebrates. *Proceedings of the Royal Society of London B: Biological Sciences.* rspb20091845. DOI :10.1098/rspb.2009.1845
- Benton M J, Dunhill A M, Lloyd G T, Marx F G, (2011) Assessing the quality of the fossil record: insights from vertebrates. In: McGowan, A.J., Smith, A.B. (Eds.), *Comparing the geological and fossil records: implications for biodiversity studies.* Geological Society, London, pp. 63–94.
- Benton M J, Ruta M, Dunhill A M, Sakamoto M (2013) The first half of tetrapod evolution, sampling proxies, and fossil record quality. *Palaeogeography, Palaeoclimatology, Palaeoecology.* 372: 18-41.
- Brocklehurst N (2015) A simulation-based examination of residual diversity estimates as a method of correcting for sampling bias. *Palaeontologia Electronica.* 18(3): 1-15
- Butler R J, Benson R B J, Carrano M T, Mannion P D, Upchurch P (2011) Sea level, dinosaur diversity and sampling biases: Investigating the ‘common cause’ hypothesis in the terrestrial realm. *Proceeding of the Royal Society of London Series B: Biological Sciences* 278(1709): 1165–1170
- Hannisdal B, Peters S E (2011) Phanerozoic Earth System evolution and marine biodiversity. *Science.* 334(1121): 1121-1124
- Hannisdal B, Haaga K A, Reitan T, Diego D, Liow L H (2016) Common species link global ecosystems to climate change. *BiorXiv.* <http://dx.doi.org/10.1101/043729>

Lane A, Janis C M, Sepkoski J J (2005) Estimating paleodiversities: a test of the taxic and phylogenetic methods. *Paleobiology*. 31(1): 21-34

Mannion P D, Upchurch P, Carrano M T, Barrett P M (2011) Testing the effect of the rock record on diversity: a multidisciplinary approach to elucidating the generic richness of sauropodomorph dinosaurs through time. *Biological Reviews*. 86(1): 157-181

Mannion P D, Benson R B J, Carrano M T, Tennant J P, Judd J, Butler R J (2015) Climate constrains the evolutionary history and biodiversity of crocodylians. *Nature Communications* 6: 8438

Martin J E, Amiot R, Lecuyer C, Benton M J (2014) Sea surface temperature contributes to marine crocodylomorph evolution. *Nature Communications* 5: 4658

Nicholson D B, Holroyd P A, Benson R B, Barrett P M (2015) Climate-mediated diversification of turtles in the Cretaceous. *Nature Communications*. 6: 1-8.

Sakamoto M, Benton M J, Venditti C (2016) Dinosaurs in decline tens of millions of years before their final extinction. *Proceedings of the National Academy of Sciences of the United States of America*. DOI: 10.1073/pnas.1521478113

Starrfelt J, Liow L H (2016) How many dinosaur species were there? Fossil bias and true richness estimated using a Poisson sampling model. *Philosophical Transactions of the Royal Society Series B: Biological Sciences*. 37(1691). DOI: 10.1098/rstb.2015.0219

Tennant J P, Mannion P D, Upchurch P (2016a) Environmental drivers of crocodyliform extinction across the Jurassic/Cretaceous transition. *Proceeding of the Royal Society of London Series B: Biological Sciences* 283(1826): 20152840

Tennant J P, Mannion P D, Upchurch P, Sutton M D Price G D (2016b) Biotic and environmental dynamics through the Late Jurassic - Early Cretaceous transition: evidence for protracted faunal and ecological turnover. *Biological Reviews*. doi:10.1111/brv.12255

Wagner P J (2000) The quality of the fossil record and the accuracy of phylogenetic inferences about sampling and diversity. *Systematic Biology*. 49(1): 65-86.

Reviewers' Comments:

Reviewer #1 (Remarks to the Author)

I am impressed by the very thorough responses and revisions, taking into account the long list of critical comments by myself and the other two reviewers. The authors acknowledge quite a few unresolved questions, but defend their approach; I am still unconvinced by the overall meaning of the study, and methods may evolve further in coming months and years to the extent that such taxon counting will be replaced by phylogenetic comparative and Bayesian modelling approaches to seek patterns, processes and drivers - but for the moment, the work is thorough and careful and in line with a number of recent papers in Nature Communications.

I recommend publication with no further revision.

MJB

Reviewer #2 (Remarks to the Author)

I appreciate the authors' effort in revising the manuscript, which now reads very nicely. I think the main point that still remains open is the lack of confidence intervals and statistical support around the estimated diversity changes and origination and extinction rates through time. Additionally, I think the authors should be more careful describing correlations between sea level changes and diversity as a causal effect, as correlations do not necessarily imply causality.

While I don't mean to necessarily criticize the SQS method, I still think that a statistical test would be important to show that the diversity patterns described here are not just the result of random nuisance. Any measure of the true amount of diversity in the deep past must be very uncertain (except for exceptionally well preserved and sampled taxa). Therefore, if the confidence intervals estimated by bootstrapping are very small (as stated by the authors in their rebuttal letter), I wonder whether this bootstrapping method is reliable. In any case, the difficulty to quantify any values of statistical support for diversity changes should be at least explicitly acknowledged. A better approach would be to simulate data sets with and without diversity changes across the J/K boundary and assess the robustness of the SQS findings.

Given the comments above, the statement "[...] representing a significant loss around the J/K transition" (p. 5 l. 145) is unwarranted.

Similarly, if the statistical significance of changes in origination and extinction rates through time is not estimated, this should be stated clearly before giving an interpretation of the results.

P. 8 l. 218-219: why are turnover rates in one interval compared to extinction rates in the previous interval? How are turnover rates calculated?

I think it would be useful to provide links to the software used in the paper (e.g. the SQS code, and the code used to estimate BC and 3T origination and extinction rates).

It might be some compatibility issue, but several of the references read as "!! INVALID CITATION !!".

Reviewer #3 (Remarks to the Author)

Dear authors,

The manuscript has been modified, sometimes in considerable depth, according to the reviewer's comments. I consider that most of my points have been addressed and have resulted in modifications of the manuscript, making it clearer and more exact. There are just a couple of points that I find still problematic:

-Line 545: marine turtles do not radiate in the earliest Cretaceous. The earliest probable chelonoid is Barremian in age and much of their radiation occurs in the Aptian-Albian interval (unless you consider estimations of cladogenesis rates based on phylogenies, but then you should do so for all other groups as well, including those where phylogeny suggest no or weak extinction at the J/K boundary).

-Lines 556-559. Again, lumping of events with the J/K transition: the main formation of the Ontong Java Plateau (not "Otong") is early Aptian, at least 20 millions years after the J/K boundary! Considering this as a potential factor suppressing recovery rates of the J/K boundary is not acceptable. Despite some efforts throughout the MS, readers without a detailed knowledge of the events of the Jurassic and the Cretaceous really have the impression that the J/K boundary concentrate a near mass-extinction-like set of biotic and physico-chemical catastrophes. This is not substantiated by the current data.

Finally, I have mentioned modification to the "manuscript" above, as the methods and analyses themselves have not been altered. I fully appreciate that applying phylogeny-informed methods has limitations and is extremely time-consuming. But my examples on ichthyosaurs were there to suggest that other groups might benefit from a critical re-evaluation of their subsampled results as well, especially when I see [lines 286-287] that observed J/K boundary survivors like *Caypullisaurus* "were excluded from our analyses as they cannot be constrained to any single time bin". I suppose cf. *Ophthalmosaurus* from the Berriasian of the UK and *Malawania* from Hauterivian-Barremian of Iraq (which all contribute to the weak ichthyosaur extinction at the J/K boundary) are also taken out of the analyses. The different time binning methods employed already show some instability in their respective results. Wouldn't you gain more confidence in your main conclusions by applying other subsampling/bias correcting methods in parallel rather than focussing on one? It is a true question; I would like to have your opinion on that.

Other than these points/questions, I consider you have adequately addressed my concerns. Best wishes and good luck,

Valentin Fischer

Rebuttal

Reviewers' comments:

Reviewer #1 (Remarks to the Author):

I am impressed by the very thorough responses and revisions, taking into account the long list of critical comments by myself and the other two reviewers. The authors acknowledge quite a few unresolved questions, but defend their approach; I am still unconvinced by the overall meaning of the study, and methods may evolve further in coming months and years to the extent that such taxon counting will be replaced by phylogenetic comparative and Bayesian modelling approaches to seek patterns, processes and drivers - but for the moment, the work is thorough and careful and in line with a number of recent papers in Nature Communications.

I recommend publication with no further revision.

MJB

Response: We thank the referee for their kind words regarding our previous changes and responses. We also hope that our modifications following concerns of the other two referees might satisfy some of their issues.

Reviewer #2 (Remarks to the Author):

I appreciate the authors' effort in revising the manuscript, which now reads very nicely. I think the main point that still remains open is the lack of confidence intervals and statistical support around the estimated diversity changes and origination and extinction rates through time. Additionally, I think the authors should be more careful describing correlations between sea level changes and diversity as a causal effect, as correlations do not necessarily imply causality.

While I don't mean to necessarily criticize the SQS method, I still think that a statistical test would be important to show that the diversity patterns described here are not just the result of random nuisance. Any measure of the true amount of diversity in the deep past must be very uncertain (except for exceptionally well preserved and sampled taxa). Therefore, if the confidence intervals estimated by bootstrapping are very small (as stated by the authors in their rebuttal letter), I wonder whether this bootstrapping method is reliable. In any case, the difficulty to quantify any values of statistical support for diversity changes should be at least explicitly acknowledged. A better approach would be to simulate data sets with and without diversity changes across the J/K boundary and assess the robustness of the SQS findings.

Response: In order to address these comments, we have followed the referee's advice and explicitly acknowledged that diversity changes are based on mean values and are not supported by additional statistical tests of robustness. We also note that this is the same for every single study that has ever employed SQS previously, including those published previously in Nature Communications: "Note that as the current application of SQS only returns mean values, we are unable to assess whether or not the resulting changes in diversity are statistically significant, as has been the case in all subsequent applications of this method."

However, we have now performed additional SQS analyses using a bootstrapping protocol, the code for which we have provided as a supplementary file. With this, curves for each of the major tetrapod clades now have 95% and 5% confidence intervals. We note that in the majority of cases, these do not change the interpretations of our results, but do serve to highlight some of the variation in estimated changes in diversity through the J/K transition.

Additionally, we have tested to see if the patterns we get from applying SQS can be broadly replicated using additional, independent measures of diversity, following suggestions of Referee #3. Recently, datasets have become available that allow us to assess phylogenetic diversity estimates for several of the groups we have analysed, including dinosaurs, ichthyosaurs, sauropterygians, and pterosaurs. We have previously performed similar comparative analyses for Crocodyliformes, but noted that this phylogenetic method suffers from biases such as incomplete lineage sampling, as mentioned in the Supplementary Methods of the present study. The results from each of our estimates of phylogenetic diversity are largely congruent with those obtained using SQS and have been incorporated into the revised version of our MS.

We agree with the referee that simulating datasets would be a welcome avenue for further research. However, doing so at the level of the present study would require a phenomenal amount of additional work. Furthermore, Alroy (2010) did perform numerous simulations to test the robustness of SQS, and provided extensive discussion of these methods in the supplementary materials for that study. As such, we feel that simulation studies are beyond the scope of the current study.

Given the comments above, the statement "[...] representing a significant loss around the J/K transition" (p. 5 l. 145) is unwarranted.

Response: We have changed this to “substantial” in order to avoid the potential confusion with a statistical assessment of patterns.

Similarly, if the statistical significance of changes in origination and extinction rates through time is not estimated, this should be stated clearly before giving an interpretation of the results.

Response: We have added the following to the relevant section in the Supplementary Methods: “As in previous studies that have used such approaches, we do not assess whether or not the changes in origination and extinction rates are statistically significant, as our analysis trials return only a single raw value.”

P. 8 l. 218-219: why are turnover rates in one interval compared to extinction rates in the previous interval? How are turnover rates calculated?

Response: We never explicitly use the term ‘turnover’, but our understanding of this is that it represents standing diversity as the product of changes in extinction and origination rates. Therefore, our interpretation here is that high extinction rates in one interval are the cause of subsequently lower diversity levels due to depletion of the species pool. “Turnover” rates are calculated based on the SQS protocol discussed in the Methods and Supplementary Methods.

I think it would be useful to provide links to the software used in the paper (e.g. the SQS code, and the code used to estimate BC and 3T origination and extinction rates).

Response: We have now attached the code used to perform the analyses as supplementary files, and additionally made them publicly available via the first author's GitHub account.

It might be some compatibility issue, but several of the references read as "!! INVALID CITATION !!!".

Response: We thank the referee for pointing this out. There was an error with our reference management software and using tracked changes, which we have corrected now. Note that this means that we have had to 'accept' several of the edited changes (moving of text) in the present version of the MS to accommodate this bug, which should still be visible in the previous version.

Reviewer #3 (Remarks to the Author):

Dear authors,

The manuscript has been modified, sometimes in considerable depth, according to the reviewer's comments. I consider that most of my points have been addressed and have resulted in modifications of the manuscript, making it clearer and more exact. There are just a couple of points that are I find still problematic:

-Line 545: marine turtles do not radiate in the earliest Cretaceous. The earliest probable chelonioid is Barremian in age and much of their radiation occurs in the Aptian-Albian interval (unless you consider estimations of cladogenesis rates based on phylogenies, but then you should do so for all other groups as well, including those where phylogeny suggest no or weak extinction at the J/K boundary).

Response: We thank the referee for pointing out this error. We meant to convey instead that the earliest Cretaceous witnesses the first occurrence times or diversifications for several other groups of semi-aquatic turtle, such as more derived eucryptodirans and paracryptodirans. We have amended the text to the following: "Ultimately, this could be related to the radiation of several important clades during the earliest Cretaceous, including birds, lissamphibians, and several groups of semi-aquatic turtles."

-Lines 556-559. Again, lumping of events with the J/K transition: the main formation of the Ontong Java Plateau (not "Otong") is early Aptian, at least 20 millions years after the J/K boundary! Considering this as a potential factor suppressing recovery rates of the J/K boundary is not acceptable. Despite some efforts throughout the MS, readers without a detailed knowledge of the events of the Jurassic and the Cretaceous really have the impression that the J/K boundary concentrate a near mass-extinction-like set of biotic and physico-chemical catastrophes. This is not substantiated by the current data.

Response: We agree with the referee and have amended the text to make this distinction clearer: "These include the Morokweng bolide impact in South Africa at the J/K boundary⁷⁰ as well as numerous episodes of Early Cretaceous flood volcanism^{6,71}, including the emergence of the Ontong Java Plateau, which was potentially a more dramatic volcanic event than that linked to the end-Cretaceous mass extinction, and might have played an important role in the evolution of tetrapods throughout the Early Cretaceous." We have also gone through the MS again to make sure that our language is clear when we are talking about events actually around the J/K boundary and those that occurred during the Early Cretaceous in general.

Finally, I have mentioned modification to the "manuscript" above, as the methods and analyses themselves have not been altered. I fully appreciate that applying phylogeny-informed methods has limitations and is extremely time-consuming. But my examples on ichthyosaurs were there to suggest that other groups might benefit from a critical re-evaluation of their subsampled results as well, especially when I see [lines 286-287] that observed J/K boundary survivors like *Caypullisaurus* "were excluded from our analyses as they cannot be constrained to any single time bin". I suppose cf. *Ophthalmosaurus* from the Berriasian of the UK and *Malawania* from Hauterivian-Barremian of Iraq (which all contribute to the weak ichthyosaur extinction at the J/K boundary) are also taken out of the analyses. The different time binning methods employed already show some instability in their respective results. Wouldn't you gain more confidence in your main conclusions by applying other subsampling/bias correcting methods in parallel rather than focussing on one? It is a true question; I would like to have your opinion on that.

Response: We agree with the referee that a parallel approach would give us more confidence in our results and, as such, have incorporated phylogenetic diversity estimates for several groups that have well-sampled global phylogenies. We have assessed phylogenetic diversity alongside SQS for pterosaurs, sauropterygians, ichthyopterygians, and the three major clades of dinosaurs. SQS and PDE data were already available for Crocodyliformes. This compliments previous studies, including the referee's own work on ichthyosaurs, by helping to look at the phylogenetic context of diversity through the J/K transition, and how this compares to results obtained using subsampling. One major advantage of comparing these approaches is that phylogenetic diversity allows for more continuous sampling of lineages, and therefore is able to provide insight into the diversity of groups during periods where sampling is sufficiently poor for SQS not to return a signal (e.g., the Berriasian). The results from each of our estimates of phylogenetic diversity largely support those obtained using SQS and have been incorporated into the revised version of our MS. We did not apply the 'residual diversity' method, as the robustness of this approach has recently been questioned, and it performs more poorly than the phylogenetic approach in simulation studies (Brocklehurst, 2015).

Regarding *Caypullisaurus*, this was actually a mistake in our text, and it was included in our analyses. Others, such as *Malawania*, were excluded from the stage level analyses, but included in the 10 million year time bin ones (within 'Cretaceous 2'). Unfortunately, poor constraints on the dating of taxa is a problem for diversity analyses. As the only Berriasian occurrences are singletons, they are excluded from our SQS trials as they do not contribute towards our understanding of the taxonomic distribution curve in a way that is useful for subsampling trials (i.e., the curve is linear). This is why we report a diversity of NA (i.e., a gap in our knowledge), instead of zero. However, our use of two different binning strategies ameliorates some of these problems, and many of our overall patterns are duplicated in the two sets of analyses. We have changed the text to the following to make it much clearer about how we have interpreted our results: "In South America we see a small decline in thalattosuchian diversity (10%), coupled with an apparent loss of all ichthyosaur and sauropterygian taxa, but we note that some poorly-dated taxa were excluded from our analyses as they cannot be constrained to any single time bin. For intervals in which sampling is too poor to produce a subsampled diversity signal, we report a result of NA (i.e., a gap in our knowledge). We acknowledge that even in these intervals there are often specimens present, but that these are singleton occurrences, or not taxonomically identifiable to the genus level." The new incorporation of phylogenetic data also helps us to identify diversity in poorly sampled time bins.

Other than these points/questions, I consider you have adequately addressed my concerns. Best wishes and good luck,

Valentin Fischer

Reviewers' Comments:

Reviewer #2 (Remarks to the Author)

Tennant and colleagues have done (again) an excellent job revising their manuscript and I look forward to seeing this study published in Nature Communications.

Best regards,
Daniele Silvestro

Reviewer #3 (Remarks to the Author)

Dear authors,

Thank you for your constructive attitude; reading and reviewing this paper has been a pleasure, because it is a job well done. I am pleased to fully support publication of this nice and thorough piece of research in Nat Coms!

Best wishes,

Valentin Fischer